# Two-colour single-molecule photoinduced electron transfer fluorescence imaging microscopy of chaperone dynamics

Jonathan Schubert [1], Andrea Schulze[1,3], Chrisostomos Prodromou [2] & Hannes Neuweiler [1✉]

Many proteins are molecular machines, whose function is dependent on multiple conformational changes that are initiated and tightly controlled through biochemical stimuli. Their mechanistic understanding calls for spectroscopy that can probe simultaneously such structural coordinates. Here we present two-colour fluorescence microscopy in combination with photoinduced electron transfer (PET) probes as a method that simultaneously detects two structural coordinates in single protein molecules, one colour per coordinate. This contrasts with the commonly applied resonance energy transfer (FRET) technique that requires two colours per coordinate. We demonstrate the technique by directly and simultaneously observing three critical structural changes within the Hsp90 molecular chaperone machinery. Our results reveal synchronicity of conformational motions at remote sites during ATPase-driven closure of the Hsp90 molecular clamp, providing evidence for a cooperativity mechanism in the chaperone's catalytic cycle. Single-molecule PET fluorescence microscopy opens up avenues in the multi-dimensional exploration of protein dynamics and allosteric mechanisms.

[1] Department of Biotechnology and Biophysics, Julius-Maximilians-University Würzburg, Am Hubland, 97074 Würzburg, Germany. [2] Biochemistry and Biomedicine, School of Life Sciences, University of Sussex, Falmer, Brighton BN1 9QG, UK. [3] Present address: Proteros Biostructures, Bunsenstr. 7a, 82152 Martinsried, Germany. ✉email: hannes.neuweiler@uni-wuerzburg.de

Living systems are organized in large networks of interacting macro-molecules that are in dynamic equilibrium. Almost all facets of biological function, like metabolism, signal transduction, or formation of structure, are accomplished by proteins that act as molecular machines. Understanding the underlying mechanisms requires the detection of multiple structural coordinates and their temporal behaviour. With the advent of cryo-electron microscopy (cryo-EM), the structural biology of large protein complexes has arrived at a level close to atomic resolution[1]. But knowledge of protein structure is only half of the information required to understand a protein's functional mechanism. A comprehensive understanding, including mechanisms of allosteric control, requires knowledge of the time order and time constants of a protein's conformational changes associated with function, acknowledging the ensemble nature of protein conformation[2–4]. Given the stochastic nature of molecular motion this is difficult to achieve using conventional spectroscopy where a vast ensemble of molecules is probed at any given time, obscuring the time order of events.

Single-molecule (sm) fluorescence resonance energy transfer (FRET) spectroscopy facilitates the direct observation of biomolecular events, probing distance changes at a scale of 2–10 nm[5–10]. FRET requires two labels, a donor and an acceptor fluorophore, to monitor a single reaction coordinate defined by their position in the biomolecule or on the reacting components. Three- and four-colour FRET detection schemes have been proposed that are capable of monitoring more than one coordinate[11–14]. Complications arise from the requirement of at least three differently coloured and site-specifically incorporated labels that, on the one hand, provide sufficient spectral overlap to facilitate FRET and, on the other, provide sufficient spectral separation to bypass cross-talk of fluorescence signals[14].

Quenching of organic fluorophores by the amino acid tryptophan (Trp) through photoinduced electron transfer (PET) detects a single protein conformational coordinate using a single label. The comparatively 'young' technique has originally been developed in the study of fast protein folding dynamics[15–17]. In aqueous solution, rhodamine and oxazine fluorophores form π–π stacking interactions with the indole side chain of Trp, which has a high oxidation potential[18]. Upon photo-excitation of the fluorophore, an electron spontaneously transfers from Trp to the fluorophore. Back transfer of an electron from the electronically excited fluorophore to Trp closes the charge transfer cycle and quenches fluorescence efficiently[15,19]. Tailored, site-directed incorporation of fluorophore and Trp into a protein translates protein conformational change along a specific coordinate into an "off" or "on" switch of fluorescence through formation or disruption of the fluorophore/Trp interaction. Contact-induced fluorescence quenching through PET can probe local conformational changes at a 1-nm scale, which complements the 2–10-nm spatial scale probed by FRET[20].

PET harbours the potential to simultaneously probe multiple conformational changes within a single protein molecule using one fluorescence colour per coordinate measured on a multi-colour setup. Here, we realized this idea. We show that a two-colour fluorescence microscope, commonly applied to probe a single coordinate using a donor and an acceptor fluorophore in a FRET setup, can be transformed into a method that simultaneously measures two conformational coordinates by channelling the two fluorescence colours into PET reporters. We exemplify the principle by the pairwise simultaneous detection of three critical structural changes of the enigmatic molecular chaperone Hsp90[21,22]. The 90-kDa heat shock protein Hsp90 transiently binds to and activates a large number of structurally and functionally diverse client proteins at their late stages of folding[23,24]. Many of the clients are involved in signal transduction. Hsp90 is a central hub of protein homeostasis with implications in disease[25]. Understanding Hsp90's chaperone mechanism could thus form the basis of new therapeutic approaches.

Hsp90 is a homo-dimeric protein, the shape of which resembles a molecular clamp[23]. A single subunit of Hsp90 consists of the nucleotide-binding N-terminal domain (NTD), separated by a charged linker from the middle domain (MD), and the C-terminal dimerization domain (CTD). The ATPase-driven catalytic cycle involves opening and closing of the molecular clamp through cycles of inter-subunit dimerization and dissociation of NTDs. The remarkably slow[21] ATPase activity of Hsp90 is rate-limited by conformational change[26,27]. The detailed mechanism by which Hsp90 activates its clients remains elusive[21]. Previous studies propose different models regarding the time order of events within the chaperone machinery[27–29]. Here, we revisit Hsp90's ATPase-driven conformational cycle applying two-colour smPET fluorescence microscopy monitoring multiple conformational changes from remote sites. We directly observe concerted motions during closure and opening of the Hsp90 molecular clamp.

## Results

**PET fluorescence reporter design and protein immobilization.** PET fluorescence reporters for protein conformational change can be designed by placing fluorophore and Trp on a protein alongside the targeted reaction coordinate[15–17]. To detect conformational motions in individual Hsp90 molecules we modified single-point cysteine (Cys, C) mutants of *yeast* Hsp90 chemically using thiol-reactive labels. *Yeast* Hsp90 is void of natural Cys that could interfere with site-specific modification. We implemented three distinct PET reporter systems introduced previously in bulk fluorescence studies[29]. The PET reporters are designed to probe three elementary, ATPase-driven structural changes of the chaperone, namely closure of the "ATP-lid" over the nucleotide binding pocket in the NTD (Lid), swapping of the N-terminal β-strands associated with inter-subunit dimerization of NTDs (domain swap, DS) and intra-subunit association of NTD and MD (NM-association) (Fig. 1a). Each conformational change of Hsp90 leads to close proximity of fluorophore and Trp, respectively, thus quenching fluorescence of the label at that site (Fig. 1a). Closure of the Lid and intra-subunit NM-association were probed using double mutants S51C-A110W and E192C-N298W, respectively. DS was probed using fluorescently modified mutant A2C on the NTD of one subunit and E162W on the NTD of the other. Previous bulk fluorescence studies show that the kinetics of conformational change probed by the PET reporters match the rate constant of ATP hydrolysis of non-modified wild-type Hsp90[29]. This indicates that the modifications do not perturb functionality of the chaperone. The previous studies further show that fluorescently modified constructs lacking engineered Trp residues exhibit no fluorescence quenching of the label at the respective site on nucleotide binding-induced conformational change[29]. This demonstrates that the observed quenching is indeed caused by interactions between dye and Trp of engineered PET reporters.

To study conformational changes in single Hsp90 molecules over extended periods of time we tethered modified proteins to glass surface supports for fluorescence imaging. To minimize fluorescent background arising from non-specific adsorption to glass, we adapted a published surface passivation and immobilization protocol[30]. The protocol involves dichlorodimethylsilane (DDS) and Tween 20 treatment for passivation. Biotinylated bovine serum albumin serves as anchor point for immobilization of the protein of interest through biotin-NeutrAvidin interaction[30] (Fig. 1b). We biotinylated fluorescently modified

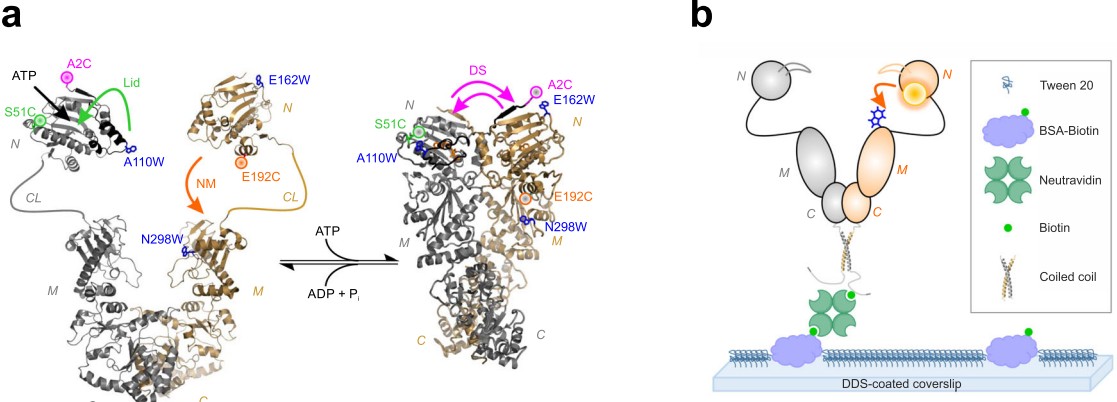

**Fig. 1 PET fluorescence reporter design and immobilization of Hsp90. a** Left: Structural model of apo Hsp90 based on crystallographic data of the NTD (pdb id 1AM1) and the MC-domain (pdb id 2CGE). NTD (N), charged linker (CL), MD (M), and CTD (C) are indicated. The ATP-binding pocket is indicated by a black arrow. Right: Crystal structure of Hsp90 in closed-clamp conformation with bound AMP-PNP (pdb id 2CG9). Mutations and fluorescence modifications (coloured spheres) constituting the PET reporters for Lid (green), NM-association (orange) and DS (magenta) are highlighted. Conformational changes initiated by binding of nucleotide and probed by PET reporters are indicated by coloured arrows. **b** Scheme of sm immobilization of hetero-dimeric Hsp90. A PET fluorescence reporter for NM-association is indicated (orange arrow).

Hsp90 molecules at a short tag introduced at the C-terminus of Hsp90, which served as a recognition sequence for a ligase. The equilibrium dissociation constant, $K_d$, of the constitutive C-terminal Hsp90 homo-dimer is 60 nM[31]. In sm fluorescence imaging experiments sub-nM concentrations of fluorescently modified protein are applied, which facilitate a sufficiently low density of spatially separate, individual proteins on the surface. In order to prevent dissociation of the Hsp90 dimer at such low concentrations, we enhanced the stability of the C-terminal dimerization interface through extension of the C-terminus by 34-residue WinZip coiled-coil sequences. WinZipA2 and Win-ZipB1 peptides are known to assemble forming leucine zipper coiled-coil hetero-dimers of high stability ($K_d = 4.5$ nM)[32]. Using C-terminal WinZipA2/WinZipB1 coiled-coils we steered hetero-dimerization of modified Hsp90 subunits containing different fluorescence labels, on the one hand, and increased Hsp90 dimer stability for sm immobilization, on the other. To minimize interference of the added coiled-coil sequences with chaperoning we inserted a flexible, eight-residue glycine-serine-rich spacer sequence after the MEEVD motif (Supplementary Fig. 1). We determined the ATPase activity of the hetero-dimeric Hsp90 construct containing the WinZipA2/WinZipB1 coiled-coil at the C-terminus and found a rate constant of $0.15 \pm 0.01$ ATP min$^{-1}$ at 25 °C (Supplementary Fig. 2). This activity was in reasonable agreement with the rate constant of $0.21 \pm 0.02$ ATP min$^{-1}$ found for wild-type Hsp90[29]. Thus, the C-terminal modifications did not significantly perturb functionality of the chaperone. We applied the WinZipA2/WinZipB1 hetero-dimer strategy to immobilize Hsp90 dimers that were site-specifically modified with either one or two fluorescence labels for the one- or two-colour smPET fluorescence imaging experiments described below.

**One-colour smPET fluorescence microscopy.** Hsp90 was modified site-specifically using the oxazine label AttoOxa11 that was previously applied in bulk fluorescence experiments[29]. AttoOxa11 together with engineered Trp served as PET fluorescence probes for Lid, DS and NM-association in one-colour smPET fluorescence detection of Hsp90 (Fig. 1). Modified Hsp90 molecules were immobilized on a glass surface as described above. We implemented sm total internal reflection fluorescence (TIRF) imaging microscopy on a home-built microscope setup. The

quality of surface passivation and specificity of the tethering reaction was confirmed in control experiments lacking NeutrAvidin, where we hardly observed fluorescent spots on glass surfaces (Supplementary Fig. 3a, b). By contrast, individual, biotin/NeutrAvidin-immobilized Hsp90 molecules were detected as bright fluorescent spots in TIRF images (Fig. 2a–c, Supplementary Fig. 3c). We used a home-built flow chamber to apply reagents to surfaces and to change solution conditions in the various experimental settings. Binding of the non-hydrolysable ATP-analogue AMP-PNP closes the Hsp90 molecular clamp[33]. Previously, PET fluorescence quenching of AttoOxa11-modified Hsp90 samples measured in the bulk showed that binding of AMP-PNP irreversibly closes the Lid, swaps the N-terminal β-strands and leads to NM-association during closure of the Hsp90 molecular clamp[29]. Here, we observed these conformational motions as abrupt transitions in sm fluorescence intensity time traces (Fig. 2a–c). In sm experiments, it is generally difficult to distinguish fluorescence quenching events from photo-physical processes or photo-bleaching. We made use of dye-sensitized photo-oxidation (DSPO) of Trp to unequivocally identify PET fluorescence quenching events in sm data. The indole side chain of Trp is readily oxidized by molecular oxygen[34]. Dyes in complex with Trp in oxygen-rich environment are known to catalyse oxidation of Trp through DSPO[35–37]. We found that the observation of long-lived conformational states associated with PET fluorescence quenching required removal of molecular oxygen. Molecular oxygen can be removed from aqueous solution through application of an enzymatic oxygen scavenger system[38]. The scavenger also prevents quenching of fluorophore excited states by oxygen, which is known to lead to blinking or photo-damage[39]. We applied the following procedure to reliably detect smPET fluorescence quenching events. We triggered conformational change through application of AMP-PNP in oxygen-depleted solution to Hsp90 sm surfaces and recorded sm fluorescence intensity time traces for durations of 10 min. In these traces, we observed off-switching of sm fluorescence. We subsequently initiated DSPO of Trp in PET reporters through application of oxygen-rich solution to the surface. We then observed fluorescence recovery of many of the PET reporters on individual Hsp90 dimers, which identified the quenching events as AMP-PNP-induced conformational changes that were associated with PET (Fig. 2a–c). The series of events is illustrated in Fig. 2d. We did not observe DSPO in previous PET fluorescence studies

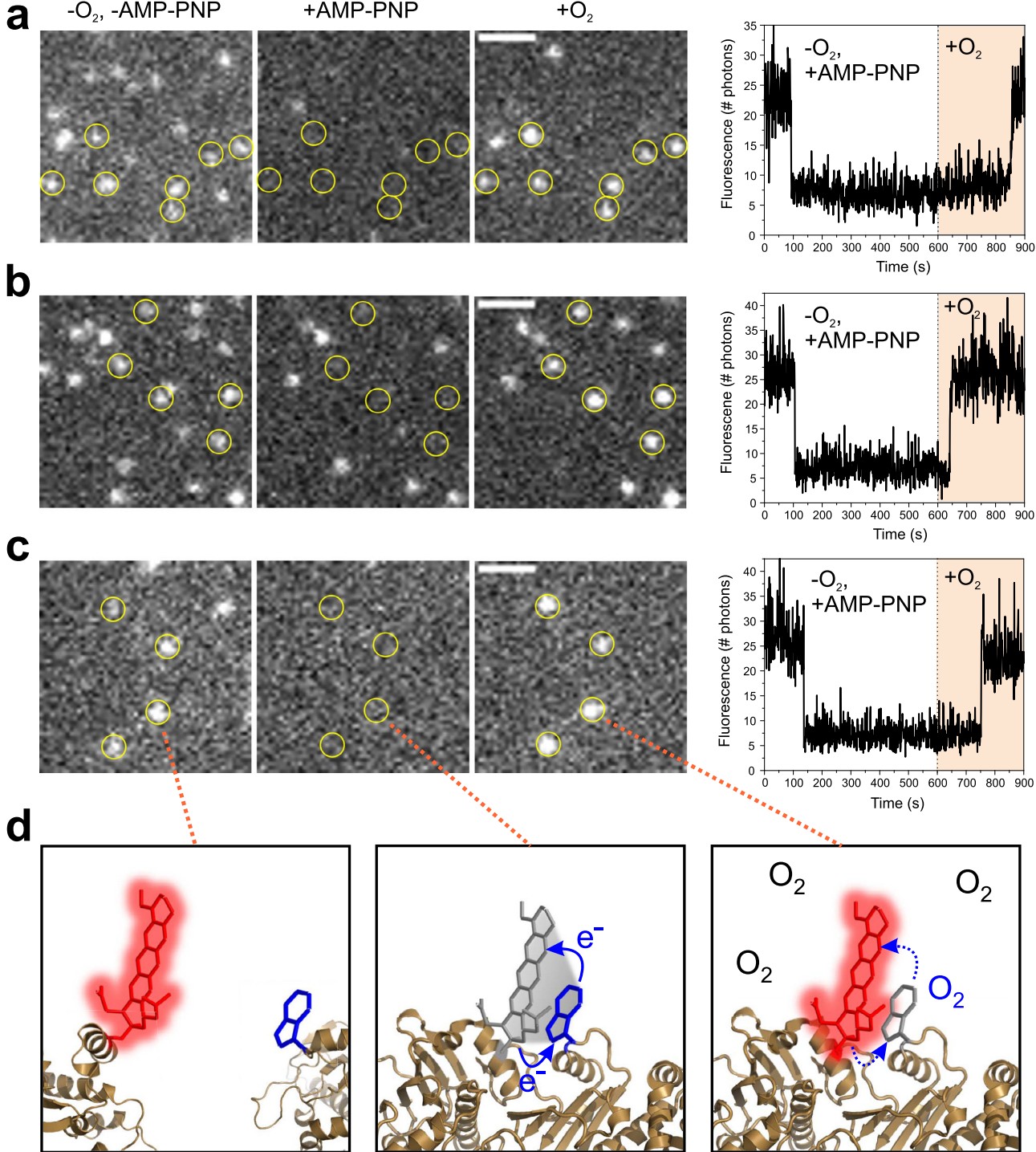

**Fig. 2 One-colour smPET fluorescence microscopy of Hsp90 dynamics.** TIRF images of single, immobilized Hsp90 molecules modified with AttoOxa11 PET fluorescence reporters for **a** DS, **b** Lid and **c** NM-association. White bars are 2-μm scale bars. For each reporter a time series of images is shown recorded in (i) oxygen-depleted solution before addition of AMP-PNP (-$O_2$, -AMP-PNP), (ii) oxygen-depleted solution after addition of AMP-PNP (+AMP-PNP), and (iii) after application of oxygen-rich solution following the addition of AMP-PNP (+$O_2$). The respective conditions are indicated on top of panel **a**. Hsp90 molecules identified to undergo PET fluorescence quenching through subsequent fluorescence recovery by DSPO are highlighted by yellow circles. Panels on the right hand side show representative fluorescence intensity time traces of individual Hsp90 molecules that undergo PET fluorescence quenching followed by fluorescence recovery through DSPO. AMP-PNP was applied at $t = 0$ s. Oxygen-rich solution was applied at $t = 600$ s. Experiments were repeated three times with similar results. Source data are provided as a Source Data file. **d** Functional principle of a PET fluorescence reporter on Hsp90 (illustrated for NM-association in a series of molecular graphics images). Conformational change results in contact between fluorophore (red sticks) and Trp (blue sticks). Fluorescence of the label is quenched through PET in oxygen-depleted solution (solid blue arrows). DSPO of Trp by molecular oxygen annihilates PET (broken blue arrows) and recovers fluorescence of the label.

carried out in bulk experiments in a cuvette in oxygen-rich solutions[29]. This is explained by the fact that excitation energies in a cuvette, using a conventional benchtop fluorimeter that has a light bulb as an excitation source, are substantially lower compared to the high local excitation energies achieved using laser excitation in TIRF microscopy. A low flux of excitation photons frustrates DSPO of Trp in PET reporters.

In conclusion, we found that local conformational changes can be explicitly detected in single Hsp90 molecules through PET fluorescence quenching using TIRF imaging microscopy. Removal of molecular oxygen is required to prevent DSPO in dye/Trp complexes, which facilitates the detection of long-lived PET fluorescence-quenched states. The discrimination of irreversible smPET events from photo-bleaching is possible through DSPO. Recovery of fluorescence of dye/Trp complexes through DSPO in oxygen-rich solution facilitates the unequivocal identification of PET events.

**Two-colour smPET fluorescence microscopy.** We expanded the method to the simultaneous detection of two conformational changes within a single protein molecule. The initial task was to find appropriate labels that could constitute two distinct PET fluorescence reporter systems within a single protein. In two-colour experiments we aimed to increase the time resolution of sm fluorescence intensity time traces in order to more accurately investigate the time order of conformational changes. To this end we applied a higher frame rate in TIRF imaging in combination with a higher laser excitation energy. The latter was required to maintain the flux of fluorescence photons. The requirements for labels suitable for two-colour smPET experiments were that they (i) showed significant quenching of fluorescence by Trp, (ii) were sufficiently bright and photo-stable to be detectable explicitly as single molecules over extended periods of time in oxygen-depleted environment, (iii) were spectrally well separate to minimize signal cross-talk and FRET, and (iv) exhibited minimal intrinsic fluorescence fluctuations during sm imaging. Under conditions of elevated laser excitation energies AttoOxa11 showed photo-physical fluctuations that could potentially interfere with detection of PET events and was therefore discarded (Fig. 3a). We screened twelve different, commercially available fluorophores for the requirements outlined above, applying a step-wise selection procedure (Supplementary Fig. 4). From this screening, two labels emerged as most suitable, namely Atto542 and Janelia Fluor 646 (JF646). Both labels fulfilled the above criteria and showed little fluctuations in sm fluorescence intensity time traces (Fig. 3b–d). To ensure that Atto542 and JF646 were capable of probing conformational changes in Hsp90 through PET, we involved them individually in a representative reporter system and performed bulk fluorescence measurements, similar as previously described using the established label AttoOxa11[29]. Closure of the molecular clamp induced by binding of AMP-PNP to fluorescently modified Hsp90, containing either Atto542 or JF646 involved in a reporter for NM-association, lead to strong quenching of fluorescence. The time course of decays followed a bi-exponential function, similar as previously described using AttoOxa11[29] (Fig. 3e, f). For both dyes the mean rate constant of NM-association, calculated from the sum of time constants of the two exponentials weighted by their amplitudes, was $0.15 \pm 0.01 \, \text{min}^{-1}$. This rate constant compared well with $0.18 \pm 0.02 \, \text{min}^{-1}$ reported previously using AttoOxa11[29]. Previous control experiments carried out on fluorescently modified Hsp90 constructs that lacked engineered Trp residues showed no quenching[29]. In further control experiments we found that the fluorescence of Atto542 and JF646 was hardly influenced by the nucleotides ATP and ADP, nor by the

aromatic amino acid L-tyrosine, which is abundantly present in *yeast* Hsp90 and could potentially form quenching interactions with labels (Supplementary Fig. 5). We concluded that Atto542 and JF646 were suitable probes for the simultaneous detection of remote, ATPase-induced conformational changes in Hsp90, using two-colour smPET fluorescence quenching.

To address the question of whether conformational changes in Hsp90 occurred sequentially or in concert we aimed for the simultaneous detection of Lid, DS and NM-association. We generated three two-colour constructs, containing both Atto542 and JF646 as pairwise PET reporters on the different structural elements covering all possible combinations of motions, namely Lid$^{\text{Atto542}}$-NM$^{\text{JF646}}$, DS$^{\text{Atto542}}$-NM$^{\text{JF646}}$ and DS$^{\text{Atto542}}$-Lid$^{\text{JF646}}$ (Fig. 4a). Atto542 and JF646 were introduced at the sites established in one-colour experiments. Site-specific two-colour modification was achieved by generating hetero-dimers of Hsp90 subunits that were modified with either Atto542 or JF646 using the WinZipA2/WinZipB1 coiled-coil strategy described above. Two-colour detection was achieved through the simultaneous excitation of Atto542 and JF646 fluorescence using a green and red laser light source. Individual, dually labelled Hsp90 molecules were identified through the observation of co-localization of Atto542 and JF646 fluorescence signals in sm TIRF images (Fig. 4b).

Binding and hydrolysis of ATP is known to drive Hsp90 repeatedly through its chaperone catalytic cycle[21,22]. We initiated conformational cycling by adding excess ATP in oxygen-depleted solutions to the immobilized Hsp90 constructs Lid$^{\text{Atto542}}$-NM$^{\text{JF646}}$, DS$^{\text{Atto542}}$-NM$^{\text{JF646}}$ and DS$^{\text{Atto542}}$-Lid$^{\text{JF646}}$. Conformational changes associated with closing and opening of the molecular clamp were observed simultaneously from the three remote sites in two-colour sm fluorescence intensity time traces. They were evident as spontaneous, stochastic events of fluorescence quenching and recovery (Fig. 4c). By contrast, sm fluorescence time traces recorded in the absence of ATP showed hardly fluorescence fluctuations (Fig. 4d). This demonstrated that the fluctuations observed in the presence of ATP originated from PET fluorescence quenching and recovery at local sites in Hsp90 during conformational cycling. The large majority of ATPase-induced fluorescence fluctuations occurred in concert in all three constructs (Fig. 4c). However, we also observed some rare non-synchronous transitions, besides eventual photo-bleaching of fluorophores (Supplementary Fig. 6). Non-synchronous transitions were also observed as rare events in some of the time traces recorded under nucleotide-free conditions (Supplementary Fig. 7). It was apparent that non-synchronous transitions occurred only from fluorescent to fluorescence-quenched states and not vice versa. In order to assess whether the non-synchronous fluctuations originated from processes other than PET we generated a control construct that did not contain engineered Trp residues, namely DS$^{\text{Atto542-noTrp}}$-NM$^{\text{JF646-noTrp}}$. Construct DS$^{\text{Atto542-noTrp}}$-NM$^{\text{JF646-noTrp}}$ was the same construct as DS$^{\text{Atto542}}$-NM$^{\text{JF646}}$ but lacked mutations E162W and N298W serving as PET electron donors in DS$^{\text{Atto542}}$-NM$^{\text{JF646}}$. We measured two-colour smPET fluorescence intensity time traces from DS$^{\text{Atto542-noTrp}}$-NM$^{\text{JF646-noTrp}}$ and found that it was void of fluctuations, both in the presence and in the absence of ATP (Supplementary Fig. 8). This indicated that the non-synchronous transitions observed from Trp-containing constructs originated from PET fluorescence quenching and thus reported on rare, spontaneous conformational transitions. Interestingly, however, we observed some rare smFRET events, i.e., anti-correlated signal changes of Atto542 and JF646 fluorescence, in the control construct DS$^{\text{Atto542-noTrp}}$-NM$^{\text{JF646-noTrp}}$ measured in the presence of ATP (Supplementary Fig. 8b). These were lacking in measurements without ATP (Supplementary Fig. 8c). The

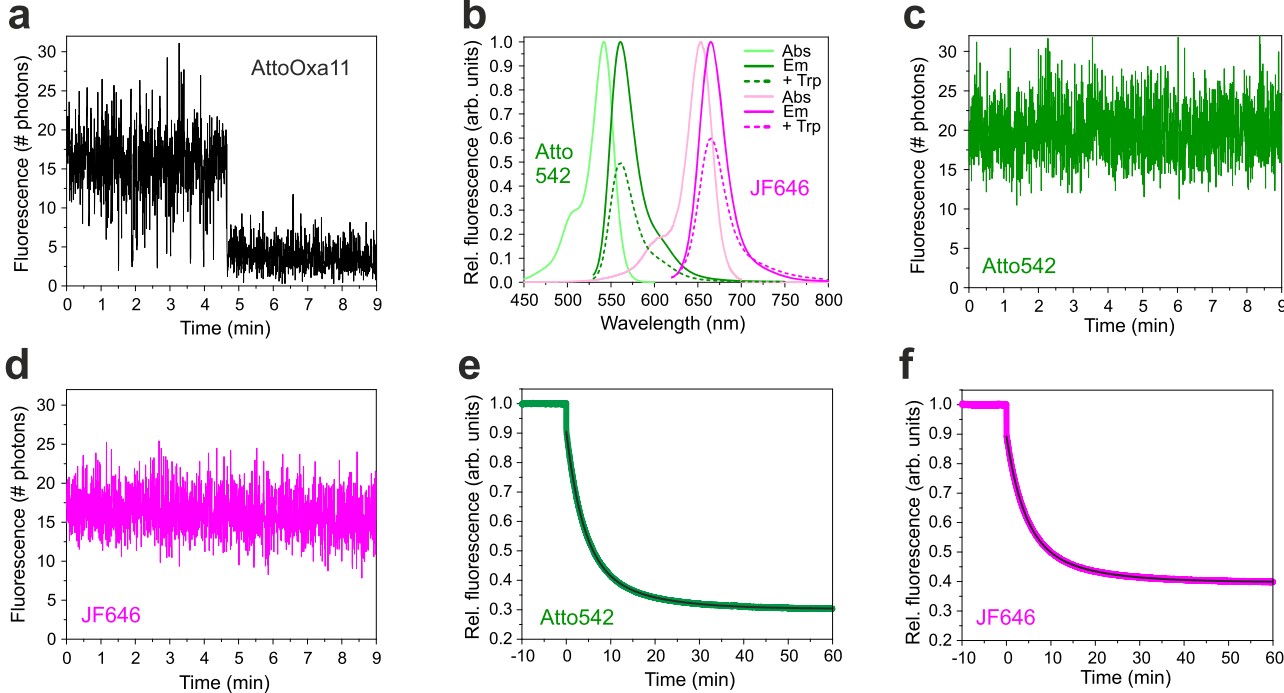

**Fig. 3 Selection of fluorophores for two-colour smPET fluorescence microscopy. a** Sm fluorescence intensity time traces of an AttoOxa11-Hsp90 conjugate immobilized on a glass surface, recorded under nucleotide-free conditions in oxygen-depleted solution. Data were measured using an integration time of 0.3 s per frame at a laser excitation energy of 10 W/cm². **b** Absorption (Abs) and fluorescence emission (Em) spectra of Atto542 and JF646 (solid lines). Emission spectra recorded in presence of 25 mM Trp are shown as dashed lines. **c, d** Sm fluorescence intensity time traces of Atto542-Hsp90 and JF646-Hsp90 conjugates immobilized on a glass surface, recorded under nucleotide-free conditions in oxygen-depleted solution. Fluorescence emission intensities were measured using an integration time of 0.3 s per frame at a laser excitation energy of 10 W/cm². **e, f** Bulk fluorescence intensities recorded from Atto542 and JF646 involved in a PET fluorescence reporter for NM-association of Hsp90 (modified mutant E192C-N298W), respectively. Measurements were carried out in cuvette experiments using a benchtop fluorimeter. 4 mM AMP-PNP was added at $t = 0$ min. Black lines are bi-exponential fits to the data. Source data are provided as a Source Data file.

fluorescence emission spectrum of Atto542 shows some overlap with the absorption spectrum of JF646, indicating that FRET from Atto542 to JF646 is possible (Fig. 3b) and explaining the observation. Two-colour time traces recorded from Hsp90 constructs containing intact PET reporters were void of FRET because PET efficiently depopulates the electronically excited states of both labels upon conformational change.

**The ATPase-driven conformational cycle of Hsp90 is dominated by concerted motions.** We quantified the numbers of synchronous ATPase-driven conformational changes of Hsp90 detected in two-colour smPET fluorescence data. For Lid$^{Atto542}$-NM$^{JF646}$, DS$^{Atto542}$-NM$^{JF646}$ and DS$^{Atto542}$-Lid$^{JF646}$ we analyzed sm fluorescence intensity time traces from about 150 molecules that were successfully labelled with both fluorophores, i.e. showed co-localization of Atto542 and JF646 fluorescence in TIRF images. We assigned fluorescence fluctuations in the green and in the red detection channel as synchronous if they occurred within a time window of 1.8 s (six frames), which is a narrow time window considering the minutes time-scale of ATPase activity of Hsp90. For the reporter pairs Lid$^{Atto542}$-NM$^{JF646}$, DS$^{Atto542}$-NM$^{JF646}$ and DS$^{Atto542}$-Lid$^{JF646}$, measured in the presence of ATP, we found a total of 422 synchronous and 62 non-synchronous transitions, 516 synchronous and 55 non-synchronous transitions, and 322 synchronous and 57 non-synchronous transitions, respectively. The relative number of non-synchronous transitions was thus between 10–18%, i.e. the majority of transitions occurred in concert (Fig. 5a). We analyzed the dwell times in conformational states. A fluorescent state was assigned to an "open" conformation and a fluorescence-quenched state was

assigned to the "closed" conformation of the respective structural element (in reference to the globally open and closed conformations of the molecular clamp). Data were analyzed using cumulative sum plot representations of dwell times. Cumulative sum plots avoid complications in variations of kinetic quantities in the analysis of common histogram representations arising from variations in bin size[40,41]. Representative cumulative sum plots including their analyses are shown in Fig. 5b–d. The entire data set including analyses is shown in Supplementary Figs. 9 and 10. The majority of cumulative sums of dwell times in open states, describing kinetics of closure, were best described by bi-exponential functions (Supplementary Fig. 9). However, some of the data fitted well to mono-exponential functions (Fig. 5b–d, Supplementary Fig. 9). Bi-exponential kinetics of closure can be explained by heterogeneity of the open-clamp conformational ensemble of Hsp90[29,42,43] from which transitions occur and by the presence of non-synchronous fluctuations contained in the data. Rate constants of closure, $k_c$, calculated as the inverse of the sum of fitted time constants in bi-exponential decays weighted by the respective amplitudes, as done in previous bulk PET fluorescence measurements[29], were between ~0.5–1.5 min⁻¹. They were similar between the pairs of reporters considering experimental error (Fig. 5f). Kinetics of opening, determined from dwell times in the PET fluorescence-quenched states, were best described by mono-exponential functions (Fig. 5b–d). Some of the data required a bi-exponential function to describe them accurately (Supplementary Fig. 10). Rate constants of opening were ~5 min⁻¹ and within error between colours and variants (Fig. 5f). Similarity of kinetics of opening are explained by a structurally well-defined closed-clamp conformation from which transitions occur. In kinetics of closure, however, we observed some significant deviations of rate constants

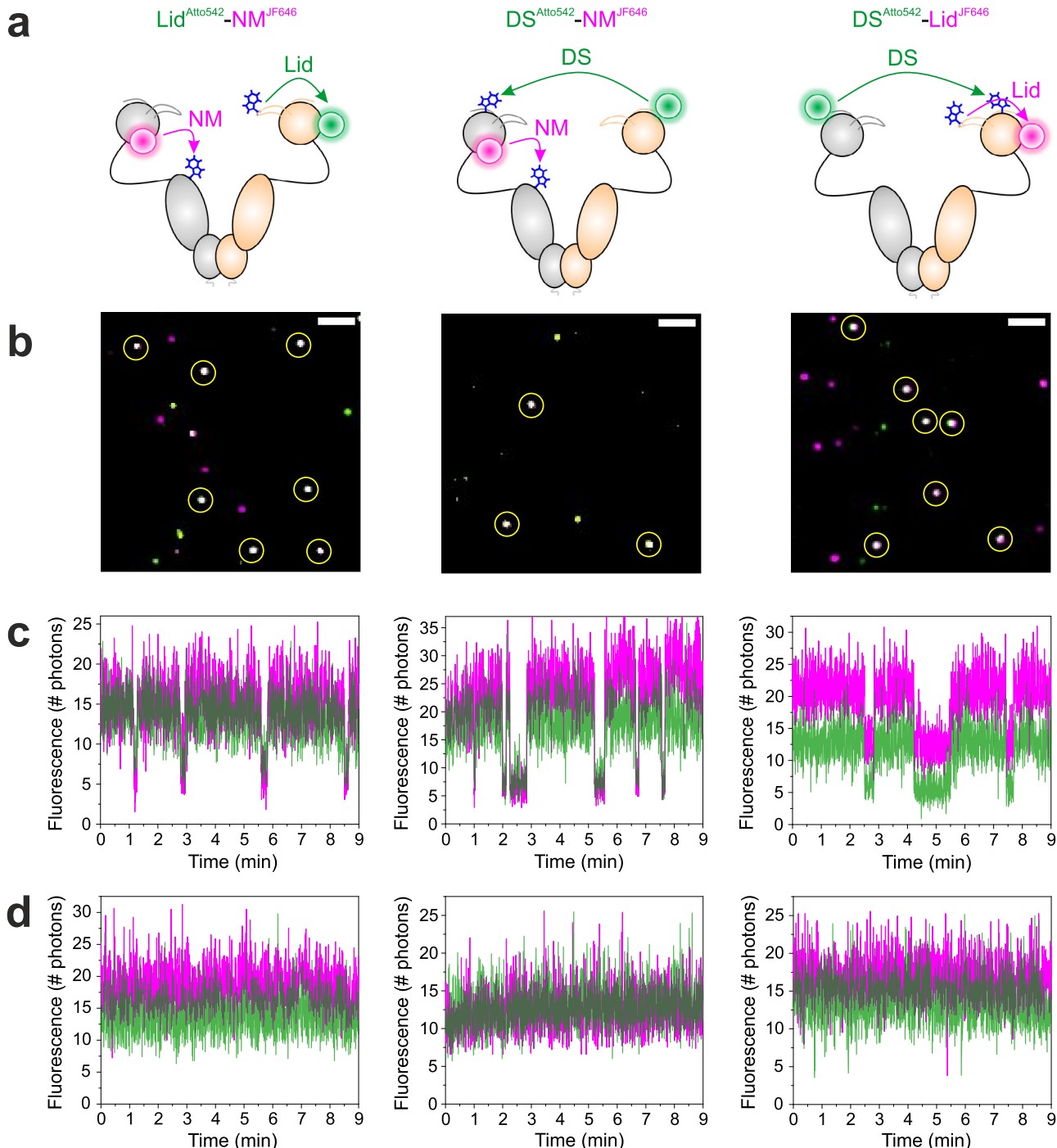

**Fig. 4 Two-colour smPET fluorescence microscopy of Hsp90 dynamics. a** Two-colour PET fluorescence reporters on Hsp90 for the simultaneous observation of Lid, DS and NM-association, illustrated as structural models. Fluorophores and Trp are indicated by coloured spheres and blue sticks. The reported conformational changes are indicated by arrows. **b** Representative two-colour smTIRF images of two-colour PET reporter-containing Hsp90 constructs tethered to glass surfaces (green: Atto542 fluorescence, magenta: JF646 fluorescence). Dually labelled Hsp90 dimers were identified by co-localization of Atto542 and JF646 fluorescence signals (white spots, highlighted by yellow circles). White bars are 1-μm scale bars. **c, d** Representative two-colour fluorescence intensity time traces recorded from individual Hsp90 molecules containing two-colour PET reporters indicated in panel **a**. Traces measured in the presence of 4 mM ATP are shown in **c**. Traces measured in the absence of ATP are shown in **d**. Panels arranged in vertical columns belong to the same construct (specified in **a**). Experiments were repeated three times with similar results. Source data are provided as a Source Data file.

between colours and constructs, which can be explained by hetero-geneity of the open-clamp conformational ensemble of Hsp90, the presence of non-synchronous fluctuations and uncertainties of fits to single-molecule data that suffer from inherently low statistics com-pared to bulk fluorescence signals[29]. Rate constants of closure of

constructs containing the DS reporter were slightly higher compared to the ones measured from Lid$^{Atto542}$-NM$^{JF646}$ (Fig. 5f). In the DS reporter, the N-terminus of Hsp90 is modified with a label. Since the N-terminal tail of Hsp90 is known to modulate its ATPase activity[44,45] we speculated that the aromatic label placed on the

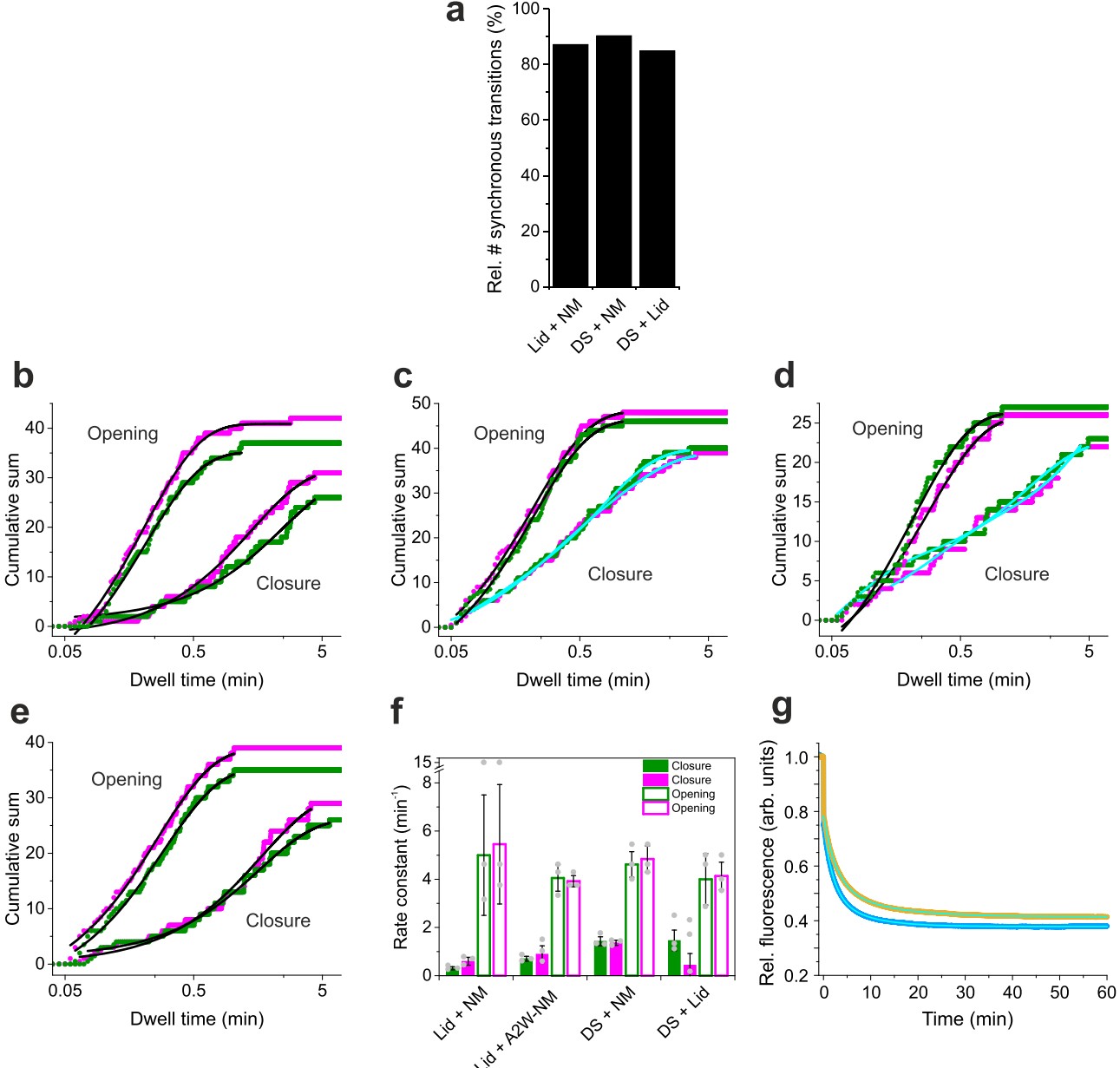

**Fig. 5 Cooperativity and kinetics of conformational cycling of Hsp90. a** Relative number of synchronous transitions detected from two-colour smPET fluorescence intensity time traces recorded from pairwise reporters for Lid, DS and NM-association. **b–e** Cumulative sum plots of dwell times in fluorescent states (kinetics of closure) and in fluorescence-quenched states (kinetics of opening) determined from the reporter pairs Lid$^{Atto542}$-NM$^{JF646}$ (**b**), DS$^{Atto542}$-NM$^{JF646}$ (**c**), DS$^{Atto542}$-Lid$^{JF646}$ (**d**) and Lid$^{Atto542}$-A2W-NM$^{JF646}$ (**e**). Data shown in green and magenta are from fluorescence of Atto542 and JF646, respectively. Black lines are mono-exponential fits to the data. Cyan lines are bi-exponential fits to the data. **f** Mean rate constants of closure (closed bars) and opening (open bars) determined from fits to the data shown in **b–e**. Values are the mean of three measurements and errors are standard deviations ($n = 3$; ±s.d.). **g** Bulk fluorescence measurements of kinetics of NM-association of Hsp90 probed by a one-colour PET reporter involving JF646 as a probe (orange). Data from the same construct but containing mutation A2W on the opposite subunit are shown in blue. Measurements were carried out on a benchtop fluorimeter. 4 mM AMP-PNP was added at $t = 0$ min. Cyan lines are bi-exponential fits to the data. Source data are provided as a Source Data file.

N-terminus at position A2C was responsible for acceleration. To test this hypothesis we introduced Trp as a natural, aromatic amino acid to the N-terminus of Hsp90 using site-directed mutagenesis (mutant A2W). We measured and analyzed kinetics of two-colour smPET fluorescence data recorded from construct Lid$^{Atto542}$-NM$^{JF646}$ containing mutation A2W (termed Lid$^{Atto542}$-A2W-NM$^{JF646}$). We found that the rate constant of closure of construct A2W was higher compared to the one determined from the construct lacking mutation A2W (Fig. 5e, f; Supplementary Table 1). We measured the

effect of mutation A2W on kinetics of closure of one-colour Hsp90 reporter NM$^{JF646}$ in bulk using a benchtop fluorimeter. The mean rate constants of NM-association, triggered by binding of AMP-PNP in bulk, was $0.18 \pm 0.01$ min$^{-1}$ forfluorescently modified Hsp90 without Trp A2W, and $0.29 \pm 0.03$ min$^{-1}$ for fluorescently modified mutant A2W, which corresponded to a 1.6-fold acceleration (Fig. 5g). ATPase assays yielded rate constants of $0.13 \pm 0.01$ min$^{-1}$ and $0.23 \pm 0.01$ min$^{-1}$, at 25 °C, for non-modified wild-type Hsp90 and mutant A2W, respectively, which corresponded to a 1.8-fold

acceleration (Supplementary Fig. 11). Results confirmed the accelerating effect induced by an aromatic moiety at the N-terminus of Hsp90 and our analysis of two-colour smPET fluorescence data.

We finally analyzed the dwell times of fluorescence-quenched states that originated from non-synchronous transitions in separation. Because the non-synchronous transitions were observed as rare events, we pooled data recorded from all three constructs (i.e. Lid[Atto542]-NM[JF646], DS[Atto542]-NM[JF646] and DS[Atto542]-Lid[JF646]) in order to obtain reasonable statistics for kinetic analysis. The resulting cumulative sum plots were well described by mono-exponential functions, both for data recorded in the presence and in the absence of ATP (Supplementary Fig. 12). The time constants derived from data fits were $10 \pm 2$ s and $9 \pm 2$ s, corresponding to rate constants of $6 \pm 1$ min$^{-1}$ and $7 \pm 1$ min$^{-1}$, for data recorded in the presence and in the absence of ATP, respectively. The kinetics were similar to the ones of opening determined from synchronous transitions (Fig. 5f, Supplementary Table 2).

## Discussion
The number of high-resolution structures of proteins and their complexes deposited in the Protein Data Bank (www.rcsb.org) continues to increase. But our understanding of protein functional mechanisms is limited by challenges in detecting the underlying dynamics[2,4]. Here, we introduced two-colour smPET fluorescence microscopy as a method that can detect two conformational coordinates in proteins simultaneously and exemplified it on the molecular chaperone Hsp90.

Hsp90 activates client proteins through an elusive ATPase-driven conformational cycle[21]. The nucleotide-free apo state is a heterogeneous ensemble of conformers where the individual domains, which are loosely connected via flexible linkers, are free to move. The conformations in the ensemble resemble beads on a string[42,43]. The myriad of conformations accessible to Hsp90 in the apo state is thought to be responsible for its remarkable capacity to recognize a plethora of structurally and functionally unrelated client proteins[46]. During chaperoning Hsp90 escapes these extended apo conformations to form a tense state, referred to as closure of the molecular clamp. Previous biophysical studies investigated this process using FRET spectroscopy. The observed multi-exponential kinetics were interpreted as arising from the formation of discrete intermediate states populated along the pathway of clamp closure[27,28]. A sequential model was proposed where the binding of ATP to the NTD leads to early folding of the lid over the nucleotide binding pocket, followed by inter-subunit dimerization of NTDs and, finally, intra-subunit association of NTD and MD[27]. Later, these local events were measured individually using bulk PET fluorescence spectroscopy and were found to exhibit similar rate constants[29]. Kinetics were further found to be similarly modulated by mutation and on co-chaperone binding. Results suggested a cooperative model of clamp closure contrasting the sequential model. Multi-exponential kinetics were also observed in PET fluorescence studies and explained by a rugged free energy landscape underlying the apo-state conformational ensemble from which transitions occur[29]. Interestingly, domain motions within the apo-state of Hsp90, probed using gold nano-spheres, were found to occur on a broad range of time scales ranging from seconds to minutes[47]. Such breadth of time scales supports the rugged free energy landscape model and suggests the presence of a number pathways to the closed-clamp conformation, impeded by kinetic traps of various depths.

The conflicting scenarios of sequential versus concerted motions during clamp closure were inferred from modelling kinetic data. It has been pointed out previously that data fitting to

model-based equations is a rather indirect assessment and not a proof for a suggested mechanism[48]. Here, using two-colour smPET fluorescence microscopy, we found evidence for the cooperative mechanism through direct observation. We directly and simultaneously observed stochastic, ATPase-driven conformational switching of the three remote structural elements in single Hsp90 molecules. We found that >80% of transitions occurred in concert (Figs. 4c and 5a). The remaining <20% non-synchronous transitions appeared as spontaneous, non-correlated motions (Supplementary Figs. 6 and 7). We interpreted these as unsuccessful attempts of Hsp90 in closing the molecular clamp. This interpretation is in agreement with the observation of attempts of clamp-closure in the apo-state conformational ensemble using gold nano-spheres and with thermally activated fluctuations detected using smFRET[47,49].

Rate constants of closure measured using two-colour smPET fluorescence were on the order of ~1 min$^{-1}$ (Fig. 5f). The values were significantly higher than ~0.2 min$^{-1}$ measured previously in solution using bulk PET fluorescence spectroscopy[29]. Similar discrepancies between bulk and sm kinetic data are reported in FRET studies[27,28]. Higher rate constants obtained from sm data may be explained by the short observation time window of only a few minutes, which is limited by photo-bleaching. A short observation time biases the detection to fast events. Bulk fluorescence emission, by contrast, can be measured over hours covering also the slow events[27,29]. An alternative explanation for accelerated clamp closure is the reduction of conformational entropy of apo Hsp90 caused by protein immobilization on a glass surface. Immobilization excludes a considerable volume space to Hsp90's apo-state conformational ensemble, which, by contrast, is fully accessible in solution. The excluded volume space reduces the entropic cost of clamp closure and thus accelerates the reaction. We found that accelerated clamp closure of constructs containing the DS reporter originated from modification of the N-terminus of Hsp90, which is known to be a regulatory element of the chaperone's ATPase activity[44,45] (Fig. 5f, g, Supplementary Fig. 11).

However, the observed variations of kinetic quantities need to be put into perspective. A two-fold increase of a rate constant $k$ (i.e. $k/k' = 2$) translates into a change of free energy of $\Delta\Delta G = -RT\ln(k/k') = 0.4$ kcal/mol, at 25 °C. This energy increment is small compared to the change of free energy associated with a chaperone's ATPase activity[50]. The free energy associated with hydrolysis of ATP by Hsp90 has been estimated to >10 kcal/mol[51].

From PET fluorescence studies a picture of the Hsp90 conformational cycle emerges. Nanosecond PET fluorescence correlation spectroscopy shows that the N-terminal β-strand and the Lid of the NTD are not rigid structures but are highly mobile, exhibiting μs reconfiguration time constants[29]. Binding of ATP to the NTD rapidly remodels the Lid, likely releasing the self-association interface early in the cycle, thus priming the NTDs for inter-subunit dimerization[29]. Closure and opening of the Hsp90 molecular clamp involves concerted motions of Lid, DS and NM-domains, as directly observed here using two-colour smPET fluorescence (Fig. 4c, Supplementary Movies 1–3). The cooperative motions of local structural elements coordinate formation of a tense, catalytically active conformation of the chaperone.

Cooperativity in allosteric mechanism is predicted by the traditional, concerted model introduced by Monod, Wyman and Changeaux (MWC model)[52]. The MWC model states that regulatory proteins are organized as homo-multimeric assemblies that exhibit symmetry properties. Cooperative transitions between distinct conformations preserve symmetry, which contrasts a sequential model[53]. The concerted model postulates that spontaneous switching between conformations is possible even in the absence of effector molecules, i.e., is a property inherent to the

protein[53]. Structure and dynamics of Hsp90 are in striking agreement with the MWC model. Hsp90 is a homo-dimer and the nucleotide-bound closed-clamp conformation shows $C_2$ rotational symmetry[33]. Concerted motions, induced by binding and hydrolysis of ATP, preserve the symmetry (Supplementary Movie 1). Spontaneous conformational switching can occur both in the presence and in the absence of ATP, as observed here (Supplementary Figs. 6 and 7) and by others[47,49]. Although asymmetry in the activation of Hsp90 by its co-chaperone Aha1 is reported[54], the cryo-EM structure of a tense, closed-clamp Hsp90:co-chaperone:client complex shows that the symmetry of the homo-dimer in its catalytically active state is preserved[24,55].

In conclusion, two-colour smPET fluorescence microscopy emerges as a method that can simultaneously detect multiple conformational motions within a single protein molecule. A common smFRET microscope, probing a single conformational coordinate, can be readily converted into a smPET setup, probing two conformational coordinates, by implementing a second laser line as excitation source and spectrally separate labels involved in PET probes. Limitations of the method are the requirement of oxygen-free solutions for the reliable detection of smPET fluorescence quenching complicating applications in live cell environments. The observation of slow events requires immobilization, which is the case in any sm fluorescence technique, because the residence time in a confocal detection volume of freely diffusing proteins is on the order of only ~1 ms. The detection of rapid, correlated conformational motions on time scales <1 ms may be realized by applying two-colour PET fluorescence detection schemes in combination with nanosecond cross-correlation spectroscopy. Extensions of the smPET technique to more than two fluorescence colours is conceivable and may aid the elucidation of mechanisms of complex protein machineries, where a number of remote structural elements conspire to create activity. We anticipate that our method opens up avenues in the multidimensional exploration of protein dynamics and allostery, one colour per coordinate.

## Methods

**Protein synthesis, mutagenesis and modification.** Engineered genes encoding full-length Hsp82 from *yeast* were cloned into a pRSET A vector (Invitrogen) using conventional restriction-digestion and ligation protocols. The genes contained a C-terminal His$_6$-tag for purification using affinity chromatography and were modified at the C-terminus using WinZipA2 (WZA2) and WinZipB1 (WZB1) coiled-coil sequences[32], which were separated from the Hsp90 C-domain by a short GS-rich spacer sequence (GSTSGSTT). A C-terminal AviTag was introduced to modify Hsp90 subunits site-specifically with biotin. The BirA Ligase is an enzyme from *E.coli* that recognizes the 15 amino-acid peptide sequence GLNDIFEAQKIEWHE, where it conjugates a biotin to the lysine residue (K)[56]. A schematic of the engineered Hsp90 gene is shown in Supplementary Fig. 1. In PET fluorescence reporter design, single-point Cys and Trp mutants were generated using the QuikChange mutagenesis protocol. The thiol side-chain of Cys served as a site-specific point of modification for thiol reactive fluorophores. *Yeast* Hsp90 is void of native Cys that could interfere with site-specific modification. We applied our previously published design of PET reporters for the Lid, NM-association and DS[29]. Double-mutants E192C-N298W and S51C-A110W within one subunit served to monitor NM-association and movement of the Lid. Mutant A2C in one subunit and E162W in the other served to monitor DS associated with inter-subunit dimerization of NTDs. All constructs were overexpressed in *Escherichia coli* C41 (DE3) cells and purified using chromatographic methods[29].

Mutants were fluorescently modified using thiol-reactive maleimide derivatives of the commercial fluorophores Atto488, Atto532, Atto542, Atto590, Atto643, Atto700 (Atto-Tec), Alexa532 (Thermo Fisher), JF646 (Tocris) and CF660R (Biotium). Labelling was carried out in 50 mM phosphate, pH 7.5, with the ionic strength adjusted to 200 mM using potassium chloride. The solutions contained a 10-fold molar excess of tris(2-carboxyethyl)phosphine (TCEP) over the protein to prevent thiol oxidation. A 5-fold molar excess of dye was applied in reactions of 2.5 h duration at 25 °C. Labelled protein was isolated from excess dye using size exclusion chromatography (SEC) applying Sephadex G-25 resin (GE Healthcare). Fluorescently labelled Hsp90 was modified enzymatically with biotin at the C-terminal Avitag using *E. coli* BirA ligase[56] following the manufacturer's protocol (Avidity). A concentration of 20–40 μM Hsp90 was reacted with 0.05 mg/ml BirA ligase in biotinylation buffer (0.05 M bicine, pH 8.3, 10 mM ATP, 10 mM magnesium acetate and 50 μM d-biotin) for 1.5 h at 30 °C. Dialysis in 50 mM phosphate, pH 7.5, with the ionic strength adjusted to 200 mM using potassium chloride, at 4 °C overnight, was applied to remove non-conjugated biotin using a 10-kDa MWCO Slide-A-Lyzer Dialysis Cassette (Thermo Fisher).

**Protein immobilization for sm detection.** Dichlorodimethlysilane (DDS) coated coverslips were prepared as described[30], but with minor modifications in the protocol. In order to obtain clean quartz glass cover slips for single-molecule immobilization we applied the following protocol. Washing was carried out three times using deionized water (H$_2$O). Next, coverslips were sonicated for 15 min in 2% aqueous Hellmanex solution (Hellma), followed by sonication in H$_2$O and acetone for 15 min, respectively. Coverslips were washed using ethanol followed by sonication in 1 M potassium hydroxide solution for 1 h. Two final sonication steps in deionized H$_2$O were applied before coverslips were dried using a nitrogen gas stream. Dried coverslips were incubated in 75 ml hexane containing 50 μl DDS (Merck), gently shaking for 1.5 h at room temperature, rinsed and sonicated twice in hexane for 1 min, followed by drying using a nitrogen stream. DDS-coated coverslips were stored at −20 °C.

Flow chambers for sm microscopy were assembled as described[57], but a piece of Parafilm was used to seal the flow chamber instead of applying a double-sided tape. The chamber was heated to 70 °C for 1 min for sealing. Using the home-built flow chamber, 0.2 mg/ml BSA-Biotin (Sigma) in 20 mM Tris buffer, 50 mM NaCl, pH 8.0 (preparation buffer) was applied to DDS-modified quartz glass coverslips for 5 min. After a washing step using the same buffer, 0.2% Tween 20 (Sigma) in preparation buffer was applied and incubated for 10 min. The surface was initially washed with preparation buffer, followed by assay buffer (50 mM phosphate, 10 mM magnesium chloride, pH 7.5, with the ionic strength adjusted to 200 mM using potassium chloride). Fluorescently modified and biotinylated Hsp90 constructs, C-terminally modified with WZA2 and WZB1, were incubated at equimolar ratio (100 nM each) in assay buffer for 1 h at 37 °C to allow formation of heterodimers. Stoichiometric amounts of NeutrAvidin (Thermofisher) were added to Hsp90 samples and incubated for 10 min at room temperature. The reaction mixture was diluted to 100–500 pM Hsp90 construct and applied to the prepared coverslip using the flow chamber.

**One-colour and two-colour smPET fluorescence microscopy.** Sm TIRF microscopy was carried out on a home-built microscope setup[58]. The setup consisted of an IX-71 inverted microscope body (Olympus) equipped with an oil-immersion objective lens (APON 60XOTIRF, Olympus) and a nosepiece stage (IX2-NPS, Olympus) for drift correction. 514-nm and a 640-nm laser lines (Genesis, MX, Coherent) were used to excite Atto542 and JF646 (or AttoOxa11) fluorescence, respectively. The excitation light was optically filtered and focused onto the back-focal plane of the objective lens. Fluorescence light was collected by the same objective, filtered by the dichroic beam splitter and band-pass filters and projected onto two electron-multiplying CCD cameras (iXon Ultra 897, Andor, Solis version 4.28.30014). Fluorescence emission was separated from excitation light using a dichroic mirror (DC5, zt405/514/635rpc, Chroma) and emission light was filtered additionally using a bandpass filter (Yokogawa 442/514/647, Semrock) inside the microscope body. The fluorescence emission path of the setup was equipped with a dual camera adapter (TuCam, Andor), splitting the fluorescence light using a long pass filter (ST, Edge Basic 635 long-pass, Semrock) and two bandpass filters (channel 1: DF 697/41, channel 2: DF 582/75, Semrock), respectively. In two-colour experiments, acquisition was performed simultaneously by directly triggering the cameras in master-slave configuration.

For sm experiments in oxygen-depleted environment, assay buffer was supplemented with 1% Glucose, 2 mM Trolox (Sigma), which is an established anti-blinking/anti-bleaching agent[59], 2 units glucose oxidase and 60 units catalase (oxygen scavenger system)[38]. The solution was incubated for 5 min to allow removal of oxygen before it was applied to the modified glass surface using the flow chamber. For one-colour experiments, 2 mM AMP-PNP in assay buffer supplemented with 1% Glucose, 2 units glucose oxidase and 60 units catalase buffer was applied. For two-colour experiments, 4 mM ATP in assay buffer, supplemented with 1% Glucose, 2 mM Trolox, 2 units glucose oxidase and 60 units catalase was applied.

Data acquisition was carried out in the TIRF illumination mode. The laser excitation intensity at the objective was measured in mW using a power meter and converted in power density by relating it to the illuminated area computed from the CCD image. The image size was $67 \times 67$ μm ($512 \times 512$ pixels, 130 nm per pixel). Acquisition of time series of TIRF images in one-colour experiments was initiated immediately after application of AMP-PNP. 600 frames were recorded using an integration time of 1 s per frame at a laser excitation energy of 5 W/cm$^2$. Acquisition of time series of TIRF images in two-colour experiments was initiated immediately after application of ATP, where 1800 frames were recorded using an integration time of 0.3 s per frame at a laser excitation energy of 10 W/cm$^2$. Slight variations in the alignment of the green and red light beams between measurements lead to slight variations in the absolute numbers of fluorescence photons detected from the probes. For alignment of Atto542 and JF646 TIRF images, TetraSpeck microspheres (Thermo Fisher), emitting light in the green and red fluorescence detection channels, were applied to the modified surfaces. TIRF images of microspheres were recorded after each final measurement of fluorescently modified Hsp90.

**ATPase assay**. ATPase activity of Hsp90 was measured using an enzyme-coupled ATPase assay involving pyruvate kinase/lactate dehydrogenase as described[60]. Activity was measured as decline of the NADH absorbance signal at the wavelength maximum of 340 nm, which is in direct stoichiometry to ADP release. Assays were carried out at 25 °C using 5 μM Hsp90 in reaction buffer (40 mM HEPES, pH 7.5, with ionic strength adjusted to 200 mM using potassium chloride), containing 0.2 mM NADH, 2 mM phosphoenol pyruvate, 50 U/ml pyruvate kinase, 50 U/ml lactate dehydrogenase, 2 mM ATP, 5 mM dithiothreitol and 10 mM magnesium chloride. The absorbance signal was measured over time using a V-650 spectro-photometer (Jasco, Spectra Manager version 2). For the formation of hetero-dimeric C-terminal WZA2/WZB1 coiled-coil Hsp90 constructs, the respective homo-dimers were mixed at equimolar ratio and incubated at 37 °C for 1 h facil-itating subunit exchange. The reaction was initiated by addition of 2 mM ATP. Background ATPase activity was tracked following inhibition of Hsp90 using 150 μM geldanamycin (Cayman Chemical).

**Bulk fluorescence experiments**. Time-dependent bulk fluorescence intensities of Hsp90 were measured in a quartz glass cuvette using a FP-6500 spectrofluorimeter (Jasco, Spectra Manager version 2). Wavelengths of fluorescence excitation/emis-sion were set to 542/562 nm or 646/668 nm, for Hsp90 samples modified with Atto542 or JF646, respectively. A Peltier thermocouple was used to adjust the sample temperature to 25 °C. Hsp90 samples were prepared in assay buffer sup-plemented with 0.3 mg/ml BSA and 0.05% Tween 20 to suppress glass surface interactions. 150 nM fluorescently modified Hsp90 was reacted with 5 μM non-labelled wild-type or mutant Hsp90 for 30 min at room temperature, prior to measurement, in order to ensure that only one subunit in dimeric Hsp90 constructs carried a fluorophore. Closure of the Hsp90 molecular clamp was initiated by adding 2 mM AMP-PNP. Fluorescence intensity time traces were fitted using a bi-exponential function.

Fluorescence quenching experiments were carried out using 150 nM fluorophore in 50 mM phosphate, pH 7.5, with the ionic strength adjusted to 200 mM using potassium chloride, and at 25 °C. The concentration of tested compounds was 25 mM each.

**Analysis of smPET fluorescence data**. Time series of recorded TIRF images were concatenated using the freely available image processing software FiJi (ImageJ, version 1.53c)[61]. Fluorescence signals from individual, modified Hsp90 molecules were localized in TIRF images using the open-source software rapi*d*STORM[62] (version 3.3). In two-colour experiments, green (Atto542) and red (JF646) sm TIRF images were aligned using elastic transformation provided by FiJi plugin BUnwarpJ[63] (version 2.6.13), aided by reference TIRF images recorded using TetraSpeck microspheres as probes (Thermo Fisher). Co-localization of green and red sm fluorescence signals in images were identified and selected as regions of interest (ROI). Within each ROI, the pixel of maximum intensity of the green and of the red detection channel was read out as a function of time using ImageJ. Transitions that occurred from fluorescent to fluorescence-quenched states, and vice versa, in the green and in the red detection channel were analyzed with regard to absolute time points at which transitions occurred and to the dwell times in the respective states, using a home-written script involving step-detection programmed in Python, version 3.8 (https://github.com/jsch111/2CsmPET). The script applies smoothing using a Gaussian filter followed by multiscale product, described and implemented by Sadler and Swami[64], to identify steps in time traces. Duration of time traces was typically 9 min. Transitions observed in the green and in the red detection channel were assigned as synchronous if they occurred within a six-frame time window (within 1.8 s). After transitions were identified, the dwell times in fluorescent and fluorescence-quenched states were extracted. Dwell times in open (fluorescent) and closed (fluorescence-quenched) states were analyzed using cumulative sum plots, which is a binning-independent method of data analysis that avoids errors in kinetics that can arise from variations in bin widths of dwell time histograms[40,41]. Cumulative sums were plotted and fitted using the data analysis software Origin 2016G. Mono-exponential and bi-exponential fits were applied to describe cumulative sums of dwell times. Mean rate constants of closure and opening, $k_c$ and $k_o$, were calculated as the inverse of the sum of time constants of bi-exponential decays weighted by the respective amplitudes, as was done pre-viously in bulk PET fluorescence studies[29]. Kinetic quantities are reported as the mean of three measurements ($n = 3$) ±s.d. of distinct samples.

**Reporting summary**. Further information on research design is available in the Nature Research Reporting Summary linked to this article.

## Data availability
The raw data underlying Figures and Supplementary Information are provided as Source Data file. The raw smTIRF images underlying Figs. 2 and 4 are provided at https://doi.org/10.5281/zenodo.5674994. Source data are provided with this paper.

## Code availability
The code used to analyze two-colour smPET fluorescence data is available at https://github.com/jsch111/2CsmPET.

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

## Acknowledgements

We thank Thomas Müller (University of Würzburg) for the kind provision of BirA Ligase. A.S. was supported by a grant of the German Excellence Initiative to the Graduate School of Life Sciences (University of Würzburg).

## Author contributions

H.N. conceptually designed the research; J.S, A.S., C.P. and H.N. designed experiments, J.S and A.S. performed experiments; J.S and A.S. analyzed the data; J.S, A.S., C.P. and H.N. wrote the paper.

## Competing interests

The authors declare no competing interests.
