## [Peer Review File · Nature Communications]

Reviewers' Comments:

Reviewer #1:

Remarks to the Author:

This manuscript from Neuweiler and colleagues demonstrates the use of multicolor PET-quenching of fluorescence to probe the conformational of HSP90 during its catalytic ATP hydrolysis cycle. HSP-90 was engineered to enable assemblage of heterodimers of HSP 90 on a glass surface that included up to two sets of fluorophore-PET pairs, wherein the fluorophores have orthogonal absorbance/emission spectra. Emission spectra were then recorded for individual heterodimers in the presence and absence of ATP. Experiments are generally well designed, and clearly establish that Trp is generally necessary for PET quenching. Experimental details are sufficiently described.

Pending the addressing of the issues/performance of the requested experiments detailed in the comments below, I would be supportive of publication of this manuscript in Nature Communications:

1. The authors mention that PET must occur when Trp and the fluorophore are in VDW contact with each other. This is not necessarily true, as PET can occur over distances. The authors cite prior literature that indicated static and dynamic quenching between Trp and Atto542, but these measurements were taken in bulk solution at very high concentrations (mM). These conditions are very different and not necessarily representative of the conditions used in this paper; thus it is not certain that Trp and fluorophore are in VDW contact. It would be safer to state close proximity (the authors mention 1 nm resolution, which is more reasonable) is required.

2. The authors report approximately 10% of observed dual color non-synchronous quenching, which was explained as photophysical fluctuations or spontaneous conformational fluctuations. These explanations were found somewhat lacking in substance. The loss of fluorescence signal clearly indicates a quenching of fluorescence somehow. What is the source of fluorescence quenching in these instances? If it were a "spontaneous" fluctuation, would this not also be observed in the -ATP experiments (the only data shown is +ATP)? This point appears rather important as 10%, whilst minor, is still a significant quantity, and potential practitioners of this technique would want to have confidence that these events are in fact noise, versus an important observation of conformational change.

To that end, the following experiments are requested to help elucidate non-synchronous quenching (or at the very least, to eliminate possibilities):

(1) stern-volmer studies using the two main fluorophores against other abundant and potential quenchers: (a) ATP (b) ADP (c) tyrosine (d) cysteine or glutathione.

(2) Estimation of the photo-oxidation potentials of Atto542, JF646 (e.g. are these strong enough photo-oxidants to engage multiple biological moieties in PET besides Trp).

(3) dual color emission studies in the presence of:

(a) mM concentrations of ADP.

(b) the presence of the other components of the experiment (e.g. bsa-biotin, tween 20, etc.)

4. While this is an overall elegant approach to studying HSP-90 conformational dynamics in great detail, there are clearly limitations to this method that should be actively acknowledged:

-the oxygen photosensitivity, the disconnect of studying a protein on a glass surface vs. in its native environment, and that the method, as shown here, could not be applied to a live cellular environment (compared to FRET which is readily applied to live cell environments.)

One experiment that could lend confidence to both the robustness of the method and potential to translate to more relevant environments, would be to perform an experiment in the presence of a cell lysate doped with ATP or, at a minimum, a mixture of proteins and glutathione also doped with ATP.

5. Other minor points that should be addressed:

Figure 2d: The descriptor of Trp-photooxidation is both confusing as to what it is trying to show and also not an accurate descriptor of sensitized photo-oxidation. The authors should provide a figure that accurately depicts the photooxidation process.

Figure 3b: To a non-expert it will not be clear as to which fluorophore is which based upon the current labelling scheme.

Figure 4c: Which construct corresponds to which image?

Supplementary Figure 3: What was the concentration of the fluorophore used in the pseudo-Stern-Volmer quenching screen?

Reviewer #2:

Remarks to the Author:

In a previous study Neuweiler and colleagues established photon-induced electron transfer (PET) to analyze the dynamics of the homodimeric molecular chaperone Hsp90 throughout the ATPase cycle using stopped-flow fluorescence and fluorescence correlation spectroscopy (FCS). PET is a fluorescence technique that allows to detect proximity between two proteins or two parts of the same protein. In contrast to Förster resonance energy transfer (FRET), which requires labeling with two compatible fluorophores and detects distances in the range between 2 and 10 nm, PET only requires the labeling with a single fluorophore which is quenched by the proximity of a tryptophan residue when fluorophore and tryptophan come closer than 1 nm.

In this study Neuweiler and colleagues extend their previous study in two important ways. (1) they monitored two distance changes within the homodimeric yeast Hsp90 simultaneously and (2) they immobilized Hsp90 and followed the dynamics of single Hsp90 dimers over time (smPET). In the first part of the study the authors screen a number of fluorophores for suitability for two-color smPET with respect to photostability and minimal spectral overlap to avoid FRET. Then they labeled yHsp90 single cysteine and tryptophan variants that included a C-terminal coiled-coil domain for high-affinity heterodimerization to generate stable dimers with each protomer labeled at a different position with a different dye and to immobilize them on a coverslip in a TIRF microscope. Three pairs of heterodimers allowed the authors to measure simultaneously the dynamics of pairs of two structural elements known from crystal structures to change position upon transition to the ATPase cycle. These rearrangements are closing of the ATP lid (lid), docking of the N-terminal domain onto the middle domain (NM), and swapping of the N-Terminal β -strands between the two N-terminal domains (DS). The authors found that all their probes reported on the anticipated conformational rearrangements and that, surprisingly, all rearrangements occur synchronously in all three constructs. Thus, suggesting a concerted closing of the lid, NM docking, and N-terminal β -strand swapping.

Overall, this article describes an exciting development of a two dye PET system on a single-molecule level with many potential future applications and provides unexpected insight into the molecular dynamics of the Hsp90 chaperone.

Major Comments

The manuscript is well written and clearly structured. The experimental design is very clear, concise and easy to understand, thereby making it amenable also for non-experts. The presented two-color smPET technology represents a major advancement in studying conformational rearrangements on short distances in single molecules and opens the door for future studies of many different systems. However, the surprising observation that all resolved conformational rearrangements occur synchronously is in contrast to previously published bulk measurements and requires additional experiments to support the authors interpretation and rule out alternative explanations.

Figure 4: The observation that all conformational rearrangements in Hsp82 take place synchronously is very surprising. An alternative explanation for this observation could be that the sampling time in the recorded TIRF movies was too short to temporally separate these rearrangements. Can the authors comment on this possibility and if possible, repeat the experiments at the smallest possible time interval to make sure they can exclude this possibility?

Figure 5 B, D, E and G: The authors should comment on their reasoning for the data points they include in their fitting. In all three panels some fits do not start at 0 and a significant amount of data points in the steady-state regime are omitted. It could very well be, that once included, these data points no longer justify the mono-exponential fitting the authors have used and therefore change the interpretation of the data. Have the authors for example tried to fit the fluorescence time-traces with a bi-exponential fits including all data points?

Figure 5F: The authors observe an influence of the labelling position on the kinetics of the actual rearrangement, in particular they found that when the A2 in Hsp82 is modified with the bulky amino acid tryptophan, conformational rearrangements are sped up. To rule out that this is also the case for other rearrangements, the authors should measure rates for the mono-color PET probes in their single molecule TIRF setup and compare the obtained time constants with the ones from the two-color PET probes. It could very well be that one PET probes influences the kinetics of the other.

Minor Comments

Figure 4C right panel: The authors observe an offset in the fluorescence (# photons) between the green and magenta trace for the DSAto542 lidJF646 PET probe that was not observed in other panels. In particular, in the middle panel a similar PET probe (DSAto542 NMJF646, again in green) is used as well and doesn't display this off-set. Can the authors comment on this off-set?

Line 132: The author state that they have performed ATPase assays on the hetero-dimeric Hsp90 that was modified with the WinZipA2 and WinZipB1. In addition, the Material and Methods also contains a section for ATPase assays. However, no such data is found in the manuscript. Can the authors please add these data? Furthermore, can the authors perform ATPase assays on the A2W Hsp82 mutant as well? This would allow the authors to connect the altered conformational dynamics of this particular variant to the enzymatic activity of the protein. The authors should also perform ATPase assays on all labelled single cysteine and single tryptophan mutants to rule out that introduction of either amino acid and labelling in case of the cysteine, will affect the function of the protein.

Line 321: The time constants of the closing times are swapped for "lidAtto542-NMJF646" and "DSAto542-NMJF646 and DSAto542-lidJF646".

Reviewer #3:

Remarks to the Author:

In their work, Schubert et al. investigate concerted motions in the chaperone Hsp90 using photo-induced electron transfer (PET) from Trp to a chromophore. The work builds on a previous study of the authors in which they studied the motions of Hsp90 upon binding of ATP (and analogs) using ensemble and FCS methods (ref. 27). Whereas the past study used constructs with single chromophore/Trp pairs to monitor the motions of separate segments of Hsp90, the authors now devised a new and elegant approach. Instead of a single Dye/Trp pair, the authors use two spectrally different Dye/Trp pairs in the same Hsp90 dimer, which allows them to directly probe concerted motions within the same molecule. To realize this idea, the authors first performed an extensive screening to find an optimal dye combination. The authors identified two dyes (Atto542 and JF646) that are spectrally sufficiently different in their excitation and emission properties, exhibit stable fluorescence emission under continuous illumination without spontaneous photo-physical fluctuations, and which are efficiently quenched by Trp. In a second step, the authors performed single molecule TIRF imaging experiments on immobilized Hsp90 dimers that contain C-terminal fusions of a WinZip domains to stabilize the dimer even at the very low concentrations used in the single molecule experiments. Using three constructs with two reporter pairs each, the authors find spontaneous open-close motions that are highly correlated within each construct. Yet, 10 – 15% of the open-close switches are asynchronous, which the authors interpret with either failed closing attempts or with some still unknown photo-physical processes. A dwell-time analysis shows that two out of three constructs show faster closure kinetics, an effect that is shown to result from an N-terminal modification of Hsp90 with a dye in these constructs.

I am very enthusiastic about this paper because the method of two dye/Trp pairs to monitor two reaction coordinates simultaneously in one protein seems to work extremely well. Unfortunately, the current form of the paper is very method-oriented and, in its present form, would qualify for a more specialized journal. Yet, I strongly believe that the authors did not take full advantage of all information in the experiments and I think with a more in-depth analysis, the work would provide important insights into the closing mechanism of Hsp90 and qualify for publication in Nature Communications. The authors should address the points detailed below.

My major criticism concerns the analysis of the trajectories. I find it quite intriguing that 80% of the transitions are concerted. This certainly suggests that conformational transitions are highly cooperative. However, the authors also find 20% of non-correlated transitions. Given their intensive screen for an optimal dye combination with stable emission, I am surprised that the authors consider photo-physics a reasonable explanation for the asynchronous transitions. Have switches to dark states been observed in constructs without Trp? If not, the alternative explanation of unsuccessful attempts in fully closing the machine seems more reasonable to me, particularly in light of the multi-exponential decays measured in bulk experiments.

In light of the above point, I suggest a more detailed analysis of the data using a kinetic model

that contains intermediates in which only parts of the chaperone can close. Hidden-Markov-Models have frequently and successfully been used for such tasks and I would have expected to see a similar analysis. This analysis can even be performed globally for the constructs DS+NM and DS+Lid even if a chromophore in position A2 artificially accelerates transitions in this variant.

The light intensity in the experiments is rather low ($10\text{W}/\text{cm}^2$ - depends a bit on how exactly the power density is computed, see my last point). I guess that the chance of both dyes being simultaneously in the excited state is rather low at this power even if both dyes are excited simultaneously (excited-state lifetimes are nanoseconds). I therefore wonder how much of the intensity changes are due to FRET from Atto542 to JF646? If Atto542 is in the excited state (S_1) and JF646 is in the ground state (S_0), this could be a significant contribution. What is the Forster-distance of this dye pair and how does it compare to the distances in Hsp90? Do the authors observe concerted transitions between both colours even in the absence of Trp?

Combined with the fast motions described in a previous publication of the authors (ref. 27), can a kinetic model be devised that would bridge timescales from milliseconds to minutes?

I missed a critical discussion of the results in light of an extensive literature, particularly the work of the Hugel & Buchner labs.

On p. 15 line 316, I didn't understand what the authors mean with the statement "The time constants of opening were more homogeneous compared to the ones for closing,...". Homogeneous is meant between the variants or between the colors? The normalized variance (normalized by the mean, Fig. 5f) seems to be similar for opening and closing.

Are the asynchronous transitions included in the kinetics of Fig. 5 b-d? If not, why do the cumulative dwell-time histograms differ between the colors? If they were included, why are the cumulative dwell-time histograms of closing still mono-exponential? Based on Fig. S4, the asynchronous excursions seem very short-lived, which should cause a fast phase with an amplitude of 10 - 20%.

Were asynchronous transitions only observed from the open to the closed state? A detailed analysis of these statistics is missing.

There seems to be a mistake on p. 17 line 321. The authors say: "... time constants of closure measured from lid-NM were significantly higher (50 - 60s) than the ones from DS-NM and DS-lid (90 - 140s)". First, what do the authors mean with time constants? Do they mean kinetic rates or relaxation times (inverse rates)? Second, based on Fig. 5 b-d and Fig. 5f, it seems that closure in the lid-NM variant is slower than in the DS-NM and DS-lid, which is also consistent with the following text. Did the authors mix the numbers in brackets (50 - 60s) and (90 - 140s) up since 50 - 60s is not slower than 90 - 140s.

I understand the idea behind the experiments in Fig. 2. Yet, all these experiments were performed with the dye Atto-Oxa11 and not with Atto542 and JF646. Although the idea of using DSPO to identify whether emission loss is due to identify PET-quenched states is cute, I wonder whether the authors used the same procedure in their dual-color experiments. If they did not, I don't see how these experiments add to the story about the conformational transitions in Hsp90.

The authors write that they use a power of $10\text{W}/\text{cm}^2$ for data acquisition, yet it is unclear how this number had been calculated. Also, a few other details are missing in the Methods part, e.g., what is the image size and what is the pixel size (in the image).

Response to the referee comments on NCOMMS-21-18674

We thank all three referees for their time and effort reviewing our manuscript and for their valuable and constructive comments. In the following, we address the comments (cited in *italics*) point by point and highlight changes made in the revised manuscript.

Reviewer 1:

Comment:

This manuscript from Neuweiler and colleagues demonstrates the use of multicolor PET-quenching of fluorescence to probe the conformational of HSP90 during its catalytic ATP hydrolysis cycle. HSP-90 was engineered to enable assemblage of heterodimers of HSP 90 on a glass surface that included up to two sets of fluorophore-PET pairs, wherein the fluorophores have orthogonal absorbance/emission spectra. Emission spectra were then recorded for individual heterodimers in the presence and absence of ATP. Experiments are generally well designed, and clearly establish that Trp is generally necessary for PET quenching. Experimental details are sufficiently described.

Pending the addressing of the issues/performance of the requested experiments detailed in the comments below, I would be supportive of publication of this manuscript in Nature Communications:

1. The authors mention that PET must occur when Trp and the fluorophore are in VDW contact with each other. This is not necessarily true, as PET can occur over distances. The authors cite prior literature that indicated static and dynamic quenching between Trp and Atto542, but these measurements were taken in bulk solution at very high concentrations (mM). These conditions are very different and not necessarily representative of the conditions used in this paper; thus it is not certain that Trp and fluorophore are in VDW contact. It would be safer to state close proximity (the authors mention 1 nm resolution, which is more reasonable) is required.

Response:

We agree with the reviewer that bi-molecular fluorescence quenching experiments involving mM concentrations of Trp do not accurately emulate interaction geometries of dye and Trp in modified protein constructs. Dye/Trp interactions in protein conjugates are modulated by the micro-environment of dye and Trp in the structure at their specific positions and by protein conformational change. It is thus safer to say that dye and Trp are in close proximity (<1 nm separation) in the fluorescence-quenched conformation, as suggested by the reviewer. We replaced “van der Waals contact” by “close proximity” on page 5 of the revised manuscript. It is also true that PET may occur over distances. Our detailed analysis of inter-molecular dye/Trp fluorescence-quenching interactions, using a combined computational/experimental approach, shows stacking interaction geometries of dye and Trp dominating in aqueous solution (ref. 18 in the revised manuscript). However, intra-molecular interactions of dye and Trp in protein conjugates could still exhibit alternative geometries compared to the bi-molecular ones.

Comment:

2. The authors report approximately 10% of observed dual color non-synchronous quenching, which was explained as photophysical fluctuations or spontaneous conformational fluctuations. These explanations were found somewhat lacking in substance. The loss of fluorescence signal

clearly indicates a quenching of fluorescence somehow. What is the source of fluorescence quenching in these instances? If it were a “spontaneous” fluctuation, would this not also be observed in the -ATP experiments (the only data shown is +ATP)? This point appears rather important as 10%, whilst minor, is still a significant quantity, and potential practitioners of this technique would want to have confidence that these events are in fact noise, versus an important observation of conformational change.

To that end, the following experiments are requested to help elucidate non-synchronous quenching (or at the very least, to eliminate possibilities):

- (1) stern-volmer studies using the two main fluorophores against other abundant and potential quenchers: (a) ATP (b) ADP (c) tyrosine (d) cysteine or glutathione.*
- (2) Estimation of the photo-oxidation potentials of Atto542, JF646 (e.g. are these strong enough photo-oxidants to engage multiple biological moieties in PET besides Trp).*
- (3) dual color emission studies in the presence of:
(a) mM concentrations of ADP.
(b) the presence of the other components of the experiment (e.g. bsa-biotin, tween 20, etc.)*

Response:

We agree with the reviewer that a closer look on the non-synchronous fluctuations is required since they may contain additional important information. Reviewer 3 below also notes that the non-synchronous fluctuations require more attention, and we agree with both reviewers that an extended analysis of this subpopulation of transitions (10-18% are non-synchronous under +ATP conditions, outlined on page 18 of the revised manuscript) is required. We revisited our analysis of two-colour smPET fluorescence data and looked in more detail at the non-synchronous fluctuations in sm traces recorded both under +ATP and –ATP conditions. First, we found that they occurred only from the fluorescent conformational states to fluorescence-quenched states and not vice versa. Second, we found that there was still a significant number of non-synchronous fluctuations present in data from measurements carried out under –ATP conditions (about half of the number detected in measurements under –ATP conditions compared to detected under +ATP conditions). Third, we carried out an additional control experiment on a new two-colour construct that lacked engineered Trp residues. Data recorded from this construct showed virtually no quenching events, indicating that the ones observed with engineered Trp most likely originate from PET. However, this construct lacking engineered Trp showed some smFRET transitions in sm fluorescence intensity time traces (i.e., anti-correlated green and red fluorescence transitions). We provide additional representative sm fluorescence time traces of two-colour constructs recorded under +ATP conditions and under –ATP conditions, and from the construct lacking engineered Trp as new Supplementary Figures 6-8. Details on additional analysis of non-synchronous fluctuations, including kinetics, and on the control construct lacking engineered Trp, are outlined in our response to comments of reviewer 3 below.

To further substantiate our findings and address possible origins of non-synchronous fluctuations we carried out additional control experiments requested by the reviewer.

(1) We investigated ATP, ADP and tyrosine as potential quenchers of fluorescence of our labels. We applied 25 mM concentration of compounds over the dyes in cuvette experiments, matching the experimental conditions of quenching experiments using Trp shown in Supplementary Figure 4. We found negligible quenching of Atto542 and JF646 by the applied compounds (new Supplementary Figure 5). We refer to these additional control experiments on page 13 of the revised manuscript. We considered experiments using cysteine (Cys) or glutathione as potential quenchers as not necessary since none of these compounds or reactive groups were involved our experiments. Whilst tyrosine residues are natively inherent in the chaperone, yeast Hsp90 is void of native Cys. In our experiments, engineered single-point Cys mutants of yeast Hsp90 were modified chemically using thiol-reactive dyes, yielding thioethers. Of note, results are in agreement with previous bulk fluorescence experiments where we investigated fluorescently modified Hsp90 constructs that lack engineered Trp residues, serving as controls, which show no quenching of fluorescence upon nucleotide binding-induced closure of the molecular clamp (ref. 29, referred to on page 13 of the revised manuscript).

(2) We are unfortunately not able to report photo-oxidation potentials of Atto542 and JF646. The company Atto-Tec refuses to provide the chemical structure of Atto542. However, comparison of the spectral properties of Atto542 with those of commonly known fluorophores suggest that it belongs to the class of rhodamine dyes. For the common fluorophore rhodamine-6G, a reduction potential of -0.95 V vs SCE is reported and the Rehm-Weller formalism suggests efficient photoinduced charge separation (ref. 15 in the revised manuscript). JF646 belongs to a new class of silane fluorophores where, to the best of our knowledge, no redox potentials have been reported yet. Our fluorescence quenching experiments involving additional compounds suggested by the reviewer (Supplementary Figure 4 and new Supplementary Figure 5) show selective quenching only by Trp, which is a signature of PET.

(3) In previous studies we found that ADP is not effective in quenching fluorophores on Hsp90, as shown for reporters for the Lid, DS and NM-association (ref. 29 in the revised manuscript, Supplementary Figure 1 therein). In the present study, we found that ADP is further not effective in quenching fluorescence of the labels Atto542 and JF646 (new Supplementary Figure 5). Moreover, ADP forms at the end of the catalytic cycle as post-hydrolysis state and is located in buried position covered by the Lid within the NTD of Hsp90, where it is inaccessible to the extrinsic label. We therefore regard additional dual-colour emission studies involving ADP as not necessary. Other components, like BSA-biotin and Tween-20 are involved in surface immobilization only. The applied surface immobilization protocol was originally developed by Taekjip Ha and coworkers for sm fluorescence imaging of modified proteins. No influence of these components on fluorescence of labels was evident or reported (ref. 30 in the revised manuscript). We note that we frequently apply BSA and Tween-20 as additives in fluorescence correlation spectroscopy of fluorescently modified protein samples and never observed any influence of these compounds on fluorescence of labels (refs. 16, 17 and 29 in the revised manuscript).

Comment:

4. While this is an overall elegant approach to studying HSP-90 conformational dynamics in great detail, there are clearly limitations to this method that should be actively acknowledged:

-the oxygen photosensitivity, the disconnect of studying a protein on a glass surface vs. in its native environment, and that the method, as shown here, could not be applied to a live cellular environment (compared to FRET which is readily applied to live cell environments.) One experiment that could lend confidence to both the robustness of the method and potential to translate to more relevant environments, would be to perform an experiment in the presence of a cell lysate doped with ATP or, at a minimum, a mixture of proteins and glutathione also doped with ATP.

Response:

We agree with the reviewer regarding the limitations of the proposed method. These should be acknowledged and discussed. We added the following paragraph to the Discussion section on pages 25-26 of the revised manuscript, which also contains and outlook for future studies: “Limitations are the requirement of an oxygen-free environment for the detection of smPET fluorescence events, which complicates applications in live cell environments. Observation of slow events requires immobilization of molecules, as is the case for any sm fluorescence technique, because free diffusion of proteins through a confocal detection volume occurs typically on a fast 1-ms time scale. The detection of correlated conformational motions on time scales faster than ~1 ms in freely diffusing proteins or protein complexes will be feasible using two-colour smPET fluorescence detection in combination with nanosecond two-colour cross-correlation spectroscopy.”

We applied the well-tried and tested surface immobilization protocol for in-vitro protein single-molecule studies developed by Taekjip Ha et al., published in Nature Methods in 2014 (ref. 30 in our revised manuscript). However, we agree with the reviewer that immobilized molecules still behave differently compared to the ones freely diffusing in a cellular environment. Immobilization of Hsp90 likely restrains the open-clamp conformational ensemble, which was shown by structural studies to resemble beads on a string. We discussed this constraint on page 24 of the revised manuscript. We agree with the reviewer that aspects of a cellular environment can be emulated by the application of doped cell lysate. However, we believe that such experiments are beyond the scope of the present work that refers to previous in-vitro studies carried out in absence of cell lysate, but should be addressed in applications of the technique to follow in future studies.

Comment:

5. Other minor points that should be addressed:

Figure 2d: The descriptor of Trp-photooxidation is both confusing as to what it is trying to show and also not an accurate descriptor of sensitized photo-oxidation. The authors should provide a figure that accurately depicts the photooxidation process.

Response:

We agree with the reviewer that the third panel of Figure 2d conveys unreasonable details of the dye-sensitized photo-oxidation (DSPO) reaction of Trp by molecular oxygen without evidence. In fact, we do not know details of the mechanism of the DSPO reaction. We therefore revised the third panel of Figure 2d and provide a simplified molecular graphics image. We

deleted the reaction arrow, which indicated electron transfer from Trp to oxygen, and now show oxygen associated with the dye/Trp complex, just indicating a DSPO reaction that abolishes PET.

Comment:

Figure 3b: To a non-expert it will not be clear as to which fluorophore is which based upon the current labelling scheme.

Response:

We revised Figure 3 and labelled the graphs in Figure 3b with the fluorophore names, in order to clarify the shown data and to make the panel consistent with the others shown in this Figure.

Comment:

Figure 4c: Which construct corresponds to which image?

Response:

We thank the reviewer for spotting this unclear presentation of data. The constructs are defined on top of panel (a) and vertical rows of panels (a) to (d) correspond to the same construct. We explained this in the revised legend of Figure 4.

Comment:

Supplementary Figure 3: What was the concentration of the fluorophore used in the pseudo-Stern-Volmer quenching screen?

Response:

The concentration of the fluorophore applied in these experiments was 150 nM. We provide this information in the revised legend of Supplementary Figure 4 and in the revised Methods section of the manuscript.

Reviewer 2:

Comment:

In a previous study Neuweiler and colleagues established photon-induced electron transfer (PET) to analyze the dynamics of the homodimeric molecular chaperone Hsp90 throughout the ATPase cycle using stopped-flow fluorescence and fluorescence correlation spectroscopy (FCS). PET is a fluorescence technique that allows to detect proximity between two proteins or two parts of the same protein. In contrast to Förster resonance energy transfer (FRET), which requires labeling with two compatible fluorophores and detects distances in the range between 2 and 10 nm, PET only requires the labeling with a single fluorophore which is quenched by the proximity of a tryptophan residue when fluorophore and tryptophan come closer than 1 nm.

In this study Neuweiler and colleagues extend their previous study in two important ways. (1) they monitored two distance changes within the homodimeric yeast Hsp90 simultaneously and

(2) they immobilized Hsp90 and followed the dynamics of single Hsp90 dimers over time (smPET). In the first part of the study the authors screen a number of fluorophores for suitability for two-color smPET with respect to photostability and minimal spectral overlap to avoid FRET. Then they labeled yHsp90 single cysteine and tryptophan variants that included a C-terminal coiled-coil domain for high-affinity heterodimerization to generate stable dimers with each protomer labeled at a different position with a different dye and to immobilize them on a coverslip in a TIRF microscope. Three pairs of heterodimers allowed the authors to measure simultaneously the dynamics of pairs of two structural elements known from crystal structures to change position upon transition to the ATPase cycle. These rearrangements are closing of the ATP lid (lid), docking of the N-terminal domain onto the middle domain (NM), and swapping of the N-Terminal β -strands between the two N-terminal domains (DS). The authors found that all their probes reported on the anticipated conformational rearrangements and that, surprisingly, all rearrangements occur synchronously in all three constructs. Thus, suggesting a concerted closing of the lid, NM docking, and N-terminal β -strand swapping.

Overall, this article describes an exciting development of a two dye PET system on a single-molecule level with many potential future applications and provides unexpected insight into the molecular dynamics of the Hsp90 chaperone.

Major Comments

The manuscript is well written and clearly structured. The experimental design is very clear, concise and easy to understand, thereby making it amenable also for non-experts. The presented two-color smPET technology represents a major advancement in studying conformational rearrangements on short distances in single molecules and opens the door for future studies of many different systems. However, the surprising observation that all resolved conformational rearrangements occur synchronously is in contrast to previously published bulk measurements and requires additional experiments to support the authors interpretation and rule out alternative explanations.

Figure 4: The observation that all conformational rearrangements in Hsp82 take place synchronously is very surprising. An alternative explanation for this observation could be that the sampling time in the recorded TIRF movies was too short to temporally separate these rearrangements. Can the authors comment on this possibility and if possible, repeat the experiments at the smallest possible time interval to make sure they can exclude this possibility?

Response:

We thank the reviewer for the comprehensive summary and detailed analysis, and for this positive assessment of our work.

We mildly disagree with the reviewer's statement that "the observation that all conformational rearrangements in Hsp82 take place synchronously is very surprising" and that "the observation ... is in contrast to previously published bulk measurements...". The reviewer is correct in saying that our observations are contrast to the interpretation of previously published FRET measurements, where populated intermediates in the chaperone's conformational cycle have been proposed (ref. 27 in the revised manuscript). Synchronicity of conformational changes observed here, however, is well in line with results from previously PET fluorescence

measurements carried out in bulk, where cooperativity of conformational rearrangements was inferred from kinetic analyses (ref. 29 in the revised manuscript). Previous bulk PET fluorescence measurements yielded similar rate constants of AMP-PNP binding-induced closure of the Lid, domain swap and NM-association, as well as similarity of modulation of rate constants at each site upon point mutation or co-chaperone binding. These results suggested that the remote conformational changes were rate-limited by a common free energy barrier (i.e., the kinetic bottleneck of the reaction was the same for each probed rearrangement). That is, conformational changes occurred cooperatively (ref. 29), which is synonymous with synchronous.

In response to the question regarding the time resolution of our two-colour smTIRF measurements we note that the data were recorded at the smallest possible photon integration time of 0.3 s per frame. The lower limit of integration time of fluorescence photons was set by the correspondingly higher laser excitation power required to obtain a reasonable flux of photons for data analysis. A reasonable flux of fluorescence photons was required to unequivocally detect single-molecule fluorescence quenching events at sufficient signal-to-noise. Further elevation of laser excitation power to further increase the time resolution of our experiment was compromised by a higher probability of undesired photo-physical events and photo-bleaching. In order to assess synchronicity of events we allowed a time window of six frames within which the events had to occur. This resulted in a time resolution of about two seconds, described in the Methods section of the revised manuscript. We therefore cannot exclude the possibility of non-synchronous events on time scales faster than two seconds. However, we render this scenario unlikely. Time constants of a sequential conformational changes associated with the formation of intermediates, reported in refs. 27 and 28, are significantly higher and such sequential events should thus have been detected in our measurements.

Comment:

Figure 5 B, D, E and G: The authors should comment on their reasoning for the data points they include in their fitting. In all three panels some fits do not start at 0 and a significant amount of data points in the steady-state regime are omitted. It could very well be, that once included, these data points no longer justify the mono-exponential fitting the authors have used and therefore change the interpretation of the data. Have the authors for example tried to fit the fluorescence time-traces with a bi-exponential fits including all data points?

Response:

We thank the reviewer for bringing a suboptimal representation of data to our attention. In fact, we included all data points in fitting. Some data sets do not contain data points at dwell times close to zero. The highest cumulative sums at dwell times >5 min do not contain data points but are misleadingly graphical continuations (horizontal lines) of the highest cumulative sum extended to nine minutes. The largest dwell time observed was smaller than nine minutes. We revised the representation of data in Figure 5, truncating the x-axis at the highest dwell time interval containing data (~6 minutes). We revised our data presentation and analysis further. We applied a logarithmic scaling of the x-axes (dwell times), which facilitates a better

assessment of the quality of fits to the data, in particular at small dwell times. All data sets fitted coarsely and reasonably well to mono-exponential functions, as originally described. But, as an outcome of the revised analysis, we found that, indeed, most of the data showing kinetics of closure were better described by bi-exponential functions than by mono-exponential ones (see new Supplementary Figures 9 and 10 showing the comparison of mono- versus bi-exponential fits to all recorded data, and revised Figure 5 showing representative data thereof). The majority of cumulative sum plots of opening remained best described by mono-exponential functions. All kinetic quantities derived from data fits are tabulated in new Supplementary Tables 1 and 2. We report all kinetic quantities as rate constants in the revised manuscript (including revised Figure 5f), which is the common unit in molecular kinetics. This revision is meant to improve readability and to facilitate comparison with ATPase activities, which are commonly reported as rate constants. Rate constants k were calculated from time constants τ ($k=1/\tau$). In case of bi-exponential fits, mean rate constants were calculated from the mean of time constants of exponentials weighted by their respective relative amplitudes (as we have done previously in ref. 29 where we also found multi-exponential kinetics of clamp closure). Rate constants of closure determined within one two-colour reporter system were within error ($\text{DS}^{\text{Atto542-NM}^{\text{JF646}}}$, $\text{lid}^{\text{Atto542-A2W-NM}^{\text{JF646}}}$) or close to within error ($\text{lid}^{\text{Atto542-NM}^{\text{JF646}}}$, $\text{DS}^{\text{Atto542-lid}^{\text{JF646}}}$) (revised Figure 5f and Supplementary Table 1). Discrepancies are explained by (i) heterogeneity of the open-clamp conformational ensemble of Hsp90 (evident in structural studies, refs. 42, 43), (ii) the inherent sensitivity of single-molecule spectroscopy to detect molecular heterogeneity and (iii) uncertainties of fits to single-molecule data that suffer from inherently low statistics of events compared to conventional bulk spectroscopy.

Mono-exponential kinetics of opening is explained by a structurally distinct closed-clamp conformation from which the opening transitions occur, contrasting closing events. We thank the reviewer for pointing out the required revision of data fitting: the revised analysis underscores heterogeneity of the open-clamp conformations and homogeneity of the closed-clamp conformation, supported by structural studies. We revised our presentation of results and their discussion on pages 18-19 and page 24 of the revised manuscript.

The fluorescence transients measured in bulk shown in Figure 5g show a rapid burst phase kinetics that were faster than the time resolution of our benchtop fluorimeter, which was evident as vertical transitions at time point zero. These burst phases were therefore excluded from fitting.

Comment:

Figure 5F: The authors observe an influence of the labelling position on the kinetics of the actual rearrangement, in particular they found that when the A2 in Hsp82 is modified with the bulky amino acid tryptophan, conformational rearrangements are sped up. To rule out that this is also the case for other rearrangements, the authors should measure rates for the mono-color PET probes in their single molecule TIRF setup and compare the obtained time constants with the ones from the two-color PET probes. It could very well be that one PET probes influences the kinetics of the other.

Response:

Modification of the N-terminus of Hsp90 (DS reporter) sped up conformational rearrangements, which we could trace back to modification of this regulatory structural element by an aromatic moiety (A2W control experiment), as mentioned by the reviewer. Construct lid^{Atto542}-NM^{JF646}, containing probes on the lid and on the NM-domain, had rate constants of 0.3-0.6 min⁻¹, as determined from two-colour smPET fluorescence data (revised Figure 5f and Supplementary Table 1 in the revised manuscript). These values are not much higher than the ATPase activity of non-modified wild-type yeast Hsp90 of ~0.2 min⁻¹ measured in bulk solution experiments (ref. 29). This shows that dynamics of Hsp90 are little influenced by probes on lid and NM-domains. Previous bulk PET fluorescence experiments carried out in solution using mono-colour probes show that kinetics of conformational change that agree well with the ATPase activity of wild type Hsp90 (ref. 29). Swap of the N-terminal β -strands (DS reporter), however, behaved differently compared to lid and NM-association. The DS reporter showed an additional fast kinetic phase (Supplementary Table 1 in ref. 29). It is known that the N-terminus of Hsp90 does have an important regulatory role, modulating the chaperone's ATPase activity (refs. 44 and 45). We performed additional ATPase assays conforming the accelerating effect of A2W on the N-terminus of Hsp90 (mutant A2W of yeast Hsp90, see also our response to the reviewer's comment below).

We generally found faster kinetics in our single-molecule measurements compared to in bulk experiments. This can be explained by (i) immobilization of Hsp90, which reduces the entropic penalty of clamp closure. Surface-immobilization excludes a considerable volume space to apo Hsp90's beads-on-a-string open-clamp conformations, present in solution, which reduces the entropic cost of formation of the closed-clamp conformation and thus accelerates the process. (ii) The observation time of single molecules is limited to a few minutes by photo-bleaching of the labels. Acceleration of rate constants has also been observed in smFRET studies where surface-immobilization of fluorescently modified Hsp90 was applied (ref. 28). However, the observed discrepancies of kinetic quantities need to be put into perspective. A two-fold increase of rate constant k ($k/k' = 2$) translates into a change of free energy of $\Delta\Delta G = -RT\ln(k/k') = 0.4$ kcal/mol at 25 °C. This is a rather small energy increment compared to the change of free energy associated with the ATPase activity of chaperones (ref. 50 in the revised manuscript). The change of free energy associated with hydrolysis of ATP by Hsp90 has been estimated to be >10 kcal/mol (ref. 51 in the revised manuscript)

We discussed the revised analyses of data and additional experiments on pages 19-20 and 24 of the manuscript. In their light and in light of the considerations above we believe that additional smPET TIRF experiments carried out on the three individual mono-colour reporter systems, testing for influences of dyes, are not necessary to support the conclusions.

Comment:

Minor Comments

Figure 4C right panel: The authors observe an offset in the fluorescence (# photons) between the green and magenta trace for the DSAtto542 lidJF646 PET probe that was not observed in other panels. In particular, in the middle panel a similar PET probe (DSAtto542 NMJF646,

again in green) is used as well and doesn't display this off-set. Can the authors comment on this off-set?

Response:

The offset reflects slight differences in the absolute numbers of fluorescence photons detected in the green and red detection channels. These differences stem from slight variations in excitation energies between individual measurements and between channels. The alignment of green and red laser light beam in the TIRF setup can vary between measurements. However, the offset in absolute intensities do not affect the relevant changes of relative fluorescence intensities between fluorescent and fluorescence-quenched states. We added the explanation to the Methods section of the revised manuscript.

Comment:

Line 132: The author state that they have performed ATPase assays on the hetero-dimeric Hsp90 that was modified with the WinZipA2 and WinZipB1. In addition, the Material and Methods also contains a section for ATPase assays. However, no such data is found in the manuscript. Can the authors please add these data? Furthermore, can the authors perform ATPase assays on the A2W Hsp82 mutant as well? This would allow the authors to connect the altered conformational dynamics of this particular variant to the enzymatic activity of the protein. The authors should also perform ATPase assays on all labelled single cysteine and single tryptophan mutants to rule out that introduction of either amino acid and labelling in case of the cysteine, will affect the function of the protein.

Response:

We added the data from the ATPase assay of WinZipA2/B1 modified Hsp90 hetero-dimer as new Supplementary Figure 2 to the revised manuscript, as requested by the reviewer. We further performed additional ATPase assays on mutant A2W of Hsp82, the data of which are provided as new Supplementary Figure 11 and are referred to on page 20 of the revised manuscript. For wild-type and mutant A2W we determined activities of $0.13 \pm 0.01 \text{ min}^{-1}$ and $0.23 \pm 0.01 \text{ min}^{-1}$ at 25 °C (1.8-fold acceleration), which is in reasonable agreement with the corresponding change of rate constants of closure from $0.18 \pm 0.01 \text{ min}^{-1}$ to $0.29 \pm 0.03 \text{ min}^{-1}$ (1.6-fold acceleration), measured by the NM-reporter in bulk PET fluorescence experiments, respectively. Our previous bulk PET fluorescence experiments show that fluorescently modified cysteine/tryptophan mutants of Hsp90 exhibit rate constants of conformational change that are very similar to the ATPase activity of the non-modified wild-type chaperone, which shows that the functionality is not affected by the modifications (ref. 29). We therefore believe that additional ATPase assays carried out on individual single-point cysteine and tryptophan mutants are not necessary to support the conclusions.

Comment:

Line 321: The time constants of the closing times are swapped for "lidAtto542-NMJF646" and "DSAtto542-NMJF646 and DSAtto542-lidJF646".

Response:

We thank the reviewer for spotting this mistake and corrected it in the revised manuscript. We further converted time constants into rate constants, both in Figure 5f and in the text, to make quantities better comparable and to improve readability of the manuscript.

Reviewer 3:

Comment:

In their work, Schubert et al. investigate concerted motions in the chaperone Hsp90 using photo-induced electron transfer (PET) from Trp to a chromophore. The work builds on a previous study of the authors in which they studied the motions of Hsp90 upon binding of ATP (and analogs) using ensemble and FCS methods (ref. 27). Whereas the past study used constructs with single chromophore/Trp pairs to monitor the motions of separate segments of Hsp90, the authors now devised a new and elegant approach. Instead of a single Dye/Trp pair, the authors use two spectrally different Dye/Trp pairs in the same Hsp90 dimer, which allows them to directly probe concerted motions within the same molecule. To realize this idea, the authors first performed an extensive screening to find an optimal dye combination. The authors identified two dyes (Atto542 and JF646) that are spectrally sufficiently different in their excitation and emission properties, exhibit stable fluorescence emission under continuous illumination without spontaneous photo-physical fluctuations, and which are efficiently quenched by Trp. In a second step, the authors performed single molecule TIRF imaging experiments on immobilized Hsp90 dimers that contain C-terminal fusions of a WinZip domain to stabilize the dimer even at the very low concentrations used in the single molecule experiments. Using three constructs with two reporter pairs each, the authors find spontaneous open-close motions that are highly correlated within each construct. Yet, 10–15% of the open-close switches are asynchronous, which the authors interpret with either failed closing attempts or with some still unknown photo-physical processes. A dwell-time analysis shows that two out of three constructs show faster closure kinetics, an effect that is shown to result from an N-terminal modification of Hsp90 with a dye in these constructs.

I am very enthusiastic about this paper because the method of two dye/Trp pairs to monitor two reaction coordinates simultaneously in one protein seems to work extremely well. Unfortunately, the current form of the paper is very method-oriented and, in its present form, would qualify for a more specialized journal. Yet, I strongly believe that the authors did not take full advantage of all information in the experiments and I think with a more in-depth analysis, the work would provide important insights into the closing mechanism of Hsp90 and qualify for publication in Nature Communications. The authors should address the points detailed below.

Response:

We thank the reviewer for this positive assessment of our work.

The slant of the paper is indeed the new method that we introduce, the application of which is demonstrated by addressing an important mechanistic question in the ATPase cycle of the chaperone Hsp90. In the revised manuscript we now expanded our discussion on the biology

of Hsp90 (pages 24-25). There are examples of method-driven papers published in Nature Communications (e.g.: Ratzke et al., Four-colour FRET reveals directionality in the Hsp90 multi-component machinery, Nat Commun 2014, 5:4192; Li et al., Automatic classification and segmentation of single-molecule fluorescence time traces with deep learning, Nat Commun 2020, 11:5833; Piatkowski et al., Broadband single-molecule excitation spectroscopy, Nat Commun 2016, 7:10411). However, we agree with the reviewer that we did not take full advantage of information in our experimental data and revised the manuscript and data analysis. The extended analysis provided additional insights into the biology of Hsp90, in particular into the nature of non-synchronous events detected, as detailed below. We put more emphasis on the biology of Hsp90 in the Results and Discussion section of the revised manuscript.

Comment:

My major criticism concerns the analysis of the trajectories. I find it quite intriguing that 80% of the transitions are concerted. This certainly suggests that conformational transitions in are highly cooperative. However, the authors also find 20% of non-correlated transitions. Given their intensive screen for an optimal dye combination with stable emission, I am surprised that the authors consider photo-physics a reasonable explanation for the asynchronous transitions. Have switches to dark states been observed in constructs without Trp? If not, the alternative explanation of unsuccessful attempts in fully closing the machine seems more reasonable to me, particularly in light of the multi-exponential decays measured in bulk experiments.

In light of the above point, I suggest a more detailed analysis of the data using a kinetic model that contains intermediates in which only parts of the chaperone can close. Hidden-Markov-Models have frequently and successfully been used for such tasks and I would have expected to see a similar analysis. This analysis can even be performed globally for the constructs DS+NM and DS+Lid even if a chromophore in position A2 artificially accelerates transitions in this variant.

Response:

We agree with the reviewer that the 10-18% of non-synchronous transitions we observed in our two-colour experiments require closer inspection and analysis, in particular since they might contain interesting biology, as pointed out by the reviewer.

We re-analysed the single-molecule time traces with regard to the non-synchronous fluctuations, addressing the points raised by the reviewer here and in the comments below. We found that also the traces measured without ATP in fact do contain some non-synchronous fluctuations, however, only about half the numbers as observed in the presence of ATP. We provided additional representative time traces recorded in presence and absence of ATP as additional Supplementary Figures 6 and 7. Results indicated that they may have biological origin, i.e. reported on Hsp90 conformational transitions. We performed an additional control experiment where we measured two-colour smPET fluorescence time traces from an Hsp90 construct that lacked engineered Trp residues, as suggested by the reviewer. The construct DS^{Atto542-noTrp}-NM^{JF646-noTrp} was the same as DS^{Atto542}-NM^{JF646} but lacked mutations E162W and N298W that provided engineered Trp as PET electron donors quenching fluorescence of labels in DS^{Atto542}-NM^{JF646}. We measured two-colour smPET fluorescence intensity time traces and

found that DS^{Atto542-noTrp}-NM^{JF646-noTrp} was essentially void of fluctuations, both in presence and absence of ATP, indicating that the non-synchronous found in the regular constructs were induced by PET fluorescence quenching. Representative time traces are provided in new Supplementary Figure 8. Interestingly, however, we observed some rare FRET events in construct DS^{Atto542-noTrp}-NM^{JF646-noTrp} in presence of ATP, i.e. anti-correlated fluorescence intensity signal changes on the green and red detection channel (new Supplementary Figure 8b), which was suggested by the reviewer in his/her comment below. These were lacking in measurements without ATP. The fluorescence emission spectrum of Atto542 shows some overlap with the absorption spectrum of JF646, showing that FRET is possible (Figure 3b). Two-colour time traces recorded from Hsp90 constructs containing engineered Trp involved in PET reporters were void of FRET transitions because PET depopulates the electronically excited states of Atto542 and JF646 efficiently on conformational change thereby preventing FRET. The additional control experiments indicated that the non-synchronous fluctuations resulted from fluorophore/Trp interactions and thus likely reported on some spontaneous local conformational transitions, which occurred rarely even in the absence of nucleotide. The findings are in line with reports from Hugel et al. showing that thermally driven attempts of Hsp90 to populate the closed-clamp conformation can occur in the absence of nucleotide (ref. 49 in the revised manuscript).

We revised our kinetic analysis (see also our response to comments of reviewer 2 above).

We found that the non-synchronous events only occurred from open, fluorescent states to fluorescence-quenched states (see also our response to the reviewer's comment below). We selectively analysed the dwell times in fluorescence-quenched states of the non-synchronous transitions. Because they occurred rarely and because of the inherently poor statistics of events in single-molecule spectroscopy we pooled the data of non-synchronous events from the three reporter systems Lid^{Atto542}-NM^{JF646}, DS^{Atto542}-NM^{JF646} and DS^{Atto542}-Lid^{JF646} in order to obtain reasonable statistics for kinetic analysis. We generated cumulative sum plots of dwell times in the fluorescence-quenched states induced by non-synchronous events and found that they were well described by a mono-exponential function. This was true both for the dwell times determined from traces recorded in the presence and in the absence of ATP (new Supplementary Figure 12). From fits to the data we found that the dwell times were ~10 sec (corresponding to a rate constant of ~6 min⁻¹) and within error between measurements carried out in the presence and in the absence of ATP. Kinetics were very similar to the ones of opening that were also overall well described by mono-exponential functions (revised Figure 5f and new Supplementary Table 2). The similarity of kinetics explains why the non-synchronous events were hardly detected as additional exponential phases in the kinetic analysis of cumulative sum plots of dwell times of opening, which contain both synchronous and non-synchronous events (revised Figure 5b-e and new Supplementary Figure 10). Kinetics of closure, however, were largely best described by bi-exponential functions (revised Figure 5b-e, new Supplementary Figure 9), which we explain by the heterogeneity of the open-clamp conformational ensemble of Hsp90, evident from structural studies, and the presence of non-synchronous events as indicated by the reviewer (see our response to comments from reviewer 2 above). We revised the description of results and their discussion in accord with these findings in the Results and Discussion sections of the revised manuscript (pages 18-20 and 23-25).

We agree with the reviewer that our single-molecule data could have also been appropriately analysed using more complex models like the Markov or Hidden-Markov model. However, the application of these models is beyond our expertise. Of note, the conformational transitions detected using the smPET fluorescence are characterized by high contrast. This is due to the efficient fluorescence quenching through PET and the consequently relatively high signal changes beyond noise. We are confident that there are no hidden states in the data that remained undetected by our analysis.

Comment:

The light intensity in the experiments is rather low (10W/cm² - depends a bit on how exactly the power density is computed, see my last point). I guess that the chance of both dyes being simultaneously in the excited state is rather low at this power even if both dyes are excited simultaneously (excited-state lifetimes are nanoseconds). I therefore wonder how much of the intensity changes are due to FRET from Atto542 to JF646? If Atto542 is in the excited state (S1) and JF646 is in the ground state (S0), this could be a significant contribution. What is the Foerster-distance of this dye pair and how does it compare to the distances in Hsp90? Do the authors observe concerted transitions between both colours even in the absence of Trp?

Response:

The laser intensity exiting the objective was measured in mW using a power meter and converted in power density by relating it to the illuminated area as computed from the CCD image. We clarified this now in the Methods section of the revised manuscript on page 30. Indeed, we kept the excitation light intensity low in order to minimize photo-bleaching, thus facilitating single-molecule detection over extended periods of time. This results in a comparatively low probability that both Atto542 and JF646 are simultaneously in the electronically excited state within a single Hsp90 dimer, which should facilitate FRET from Atto542 to JF646, as pointed out by the reviewer. In our screening for fluorophores suitable for two-colour smPET fluorescence detection we took care of selecting a pair that is spectrally well separated in order to minimize interference from FRET. However, there is residual overlap between Atto542 fluorescence emission and JF646 absorption (Figure 3b). FRET should thus in principle be possible. In response to the reviewers comment we calculated the Förster distance of Atto542/JF646 to 59 Å using the online software tool provided by www.fpbase.org/fret/ (Lambert, FPbase: a community-editable fluorescent protein database, Nat Methods 2019, 16, 277-278; Wu and Brand, Resonance Energy Transfer: Methods and Applications, Anal Biochem 1994, 218, 1-13). We indeed observed some FRET events in our data recorded from the control construct that lacked engineered Trp residues (construct DS^{Atto542-noTrp}-NM^{JF646-noTrp}, see Supplementary Figure 8 in the revised manuscript and our response to the comment above). According to the crystal structure of the closed-clamp conformation of Hsp90 (PDB id 2CG9), the distance between Atto542 on position A2C (DS) and JF646 on position E192C (NM) is 44 Å. However, the FRET events observed from construct DS^{Atto542-noTrp}-NM^{JF646-noTrp} occurred similarly rare as the non-synchronous quenching events detected in two-colour smPET fluorescence time traces. No FRET events were detected in any of the construct containing intact two-colour smPET reporters because, here, the

electronically excited states of both dyes are efficiently depopulated (quenched) by PET on conformational change of Hsp90.

Comment:

Combined with the fast motions described in a previous publication of the authors (ref. 27), can a kinetic model be devised that would bridge timescales from milliseconds to minutes?

Response:

In response to the reviewer's comment we described the conformational cycle of Hsp90 as inferred from PET fluorescence experiments on pages 24-25 of the revised manuscript. We included the μ s conformational dynamics of Lid and N-terminal β -strand in the description, which we detected previously using PET-FCS (ref. 29), and the mechanistic implications on the cycle. There is currently a gap in the experimental investigation of Hsp90 dynamics on the time window between \sim 1 ms to \sim 2 s. This gap may be closed in future studies, involving stopped-flow spectroscopy as a possibly suitable technique in the study of kinetics on this time scale.

Comment:

I missed a critical discussion of the results in light of an extensive literature, particularly the work of the Hugel & Buchner labs.

Response:

We agree with the reviewer that the discussion of results in light of the biology of Hsp90 was a bit sparse in the original manuscript. We revised the Discussion section. We expanded discussions of our results, including additional references, also to the work from the Hugel and Buchner labs, with regard to: (i) the observed spontaneous, non-synchronous fluctuations, referring to work from the Hugel and Sönnichsen labs, who also found thermally activated fluctuations (refs. 47 and 49); (ii) concerted versus sequential models of conformational change in regulatory proteins, discussing models of allostery and their identification (refs. 48, 52 and 53); (iii) symmetry of Hsp90 complexes relevant to the concerted model of allostery (Buchner et al: ref. 54, Agard et al.: ref. 24, Southworth et al.: ref. 55); (iv) energetics of ATP hydrolysis by Hsp90 including work from Buchner and Kaila (refs. 50 and 51).

Comment:

On p. 15 line 316, I didn't understand what the authors mean with the statement "The time constants of opening were more homogeneous compared to the ones for closing,...". Homogenous is meant between the variants or between the colors? The normalized variance (normalized by the mean, Fig. 5f) seems to be similar for opening and closing.

Response:

We thank the reviewer for spotting this incorrect wording. Time constants (or rate constants) cannot be "homogeneous". What we meant was that the rate constants of opening were very similar between colors and variants (within error), which is explained by a structurally distinct ("homogeneous") closed clamp conformation from which the transitions occur (revised Figure 5f). We corrected this mistake on page 19 of the revised manuscript.

Comment:

Are the asynchronous transitions included in the kinetics of Fig. 5 b-d? If not, why do the cumulative dwell-time histograms differ between the colors? If they were included, why are the cumulative dwell-time histograms of closing still mono-exponential? Based on Fig. S4, the asynchronous excursions seem very short-lived, which should cause a fast phase with an amplitude of 10 – 20%.

Response:

The non-synchronous transitions were included in the kinetics shown in Figures 5b-d but their analyses required revision (see our responses to comments from reviewers 2 and 3 above, revised Figure 5 and revised Supplementary Information). Most cumulative sum plots describing kinetics of closure were in fact best described by bi-exponential functions. While the rate constants of closure are overall in good agreement between the colours, we found some deviations that are slightly beyond error for some of the constructs (see revised Figure 5f, Supplementary Figure 9 and Supplementary Table 1). The non-synchronous transitions are a reasonable explanation for some deviations found between colours. We thank the reviewer for this additional interpretation of our data, which we included now on page 19 of the revised manuscript. In response to the reviewer's comments above, we revisited the analysis of non-synchronous transitions and found that their dwell times in off-states were on a similar scale as the dwell times of off states induced by synchronous transitions (new Supplementary Figure 12). These results are described and discussed on pages 20 and 23 of the revised manuscript.

Comment:

Were asynchronous transitions only observed from the open to the closed state? A detailed analysis of these statistics is missing.

Response:

Indeed, the non-synchronous transitions were only observed from the open to the closed states. We re-analysed their numbers and dwell times in off-states in the revised manuscript (see our response to the comment above).

Comment:

There seems to be a mistake on p. 17 line 321. The authors say: "... time constants of closure measured from lid-NM were significantly higher (50 – 60s) than the ones from DS-NM and DS-lid (90 – 140s)". First, what do the authors mean with time constants? Do they mean kinetic rates or relaxation times (inverse rates)? Second, based on Fig. 5 b-d and Fig. 5f, it seems that closure in the lid-NM variant is slower than in the DS-NM and DS-lid, which is also consistent with the following text. Did the authors mix the numbers in brackets (50 – 60s) and (90 – 140s) up since 50 – 60s is not slower than 90 – 140s.

Response:

We thank the reviewer for spotting these mistakes. We corrected the mistakes in the revised manuscript. We now report all kinetic quantities uniformly as rate constants, which is common standard in molecular kinetics, in order to improve the readability of the manuscript.

Comment:

I understand the idea behind the experiments in Fig. 2. Yet, all these experiments were performed with the dye Atto-Oxa11 and not with Atto542 and JF646. Although the idea of using DSPO to identify whether emission loss is due to identify PET-quenched states is cute, I wonder whether the authors used the same procedure in their dual-color experiments. If they did not, I don't see how these experiments add to the story about the conformational transitions in Hsp90.

Response:

The experiments shown in Figure 2 and described in Results section “One-colour smPET fluorescence microscopy” delineate the development of the technique. They show that the method can also be used in the one-colour mode and, in particular, for the detection of irreversible conformational changes. We applied AttoOxa11 in these experiments because this was the fluorophore we originally used for bulk PET fluorescence experiments. We later found that AttoOxa11 was sub-optimal for two-colour experiments, as described in the section “Two-colour smPET fluorescence microscopy”. We believe that the experiments shown in Figure 2 are important to the story because they show how one can discriminate signal changes originating from conformational change (PET) and from photo-bleaching by the DSPO experiment. The DSPO experiment is useful in the observation of irreversible conformational changes where oxygen-rich solution can be applied after the biological event has happened. We believe that this information is important for scientists interested in the application of the smPET method for the observation of irreversible biology. We explained the usefulness of this experiment now more clearly on page 10 of the revised manuscript. We did not apply DSPO in two-colour smPET experiments because they were designed to detect reversible conformational cycling in presence of ATP. This contrasted the AMP-PNP-induced irreversible closure of the Hsp90 molecular clamp detected in one-colour smPET fluorescence experiments.

Comment:

The authors write that they use a power of 10W/cm² for data acquisition, yet it is unclear how this number had been calculated. Also, a few other details are missing in the Methods part, e.g., what is the image size and what is the pixel size (in the image).

Response:

The laser intensity exiting the objective was measured in mW using a power meter and converted in power density by relating it to the illuminated area as computed from the CCD image. The image size was 67x67 μm (512x512 pixels, 130 nm per pixel). We provided these details now in the revised Methods section on page 30 of the manuscript.

Reviewers' Comments:

Reviewer #1:

Remarks to the Author:

I thank the authors for their thoughtful and detailed responses to my comments. The additional quenching studies performed with individual amino acids and other reagents, as well as the non-Trp containing construct, increase confidence in the precision of the measurement and that the observation of the non-synchronous transition events are indeed conformational fluctuations of HSP90. We also appreciate the enhanced confidence that these studies give to this reviewer regarding the specificity of the Trp-fluorophore PET fluorescence quenching event.

Whilst the addition of the redox potentials of the fluorophores would have been welcome, we are reasonably satisfied on the specificity of the quenching event as reinforced by the additional experiments.

In light of these additional studies, the observation of the non-synchronous quenching events is interesting, and the additional discussion of the potential origin of these events is welcome.

At this point, my comments have been addressed and I am supportive of the publication of this manuscript in Nature Communications.

Reviewer #2:

Remarks to the Author:

The authors addressed all my criticism and added a significant amount of data and more appropriate data analysis. Thus the manuscript is greatly improved and suitable for publication. In my opinion the methodology will be appreciated by many scientists in different areas of research. The results concerning the ATPase cycle of Hsp90 are highly interesting as several conformational changes are detected at the single molecule level at the same time.

I have only a single final minor comment:

Figure 5 C and D: The fit lines for the closure are shown in gray and it is difficult to associate the fit lines with the corresponding cumulative dwell time data. It would be better to distinguish the fit lines by different colors.

Reviewer #3:

Remarks to the Author:

The authors significantly improved the manuscript, particularly the analysis of their data. Overall I am happy with the paper in its current form and I recommend publication.

Response to the referee comments on NCOMMS-21-18674A

Reviewer 1:

Comment:

I thank the authors for their thoughtful and detailed responses to my comments. The additional quenching studies performed with individual amino acids and other reagents, as well as the non-Trp containing construct, increase confidence in the precision of the measurement and that the observation of the non-synchronous transition events are indeed conformational fluctuations of HSP90. We also appreciate the enhanced confidence that these studies give to this reviewer regarding the specificity of the Trp-fluorophore PET fluorescence quenching event.

Whilst the addition of the redox potentials of the fluorophores would have been welcome, we are reasonably satisfied on the specificity of the quenching event as reinforced by the additional experiments.

In light of these additional studies, the observation of the non-synchronous quenching events is interesting, and the additional discussion of the potential origin of these events is welcome.

At this point, my comments have been addressed and I am supportive of the publication of this manuscript in Nature Communications.

Response:

We thank referee #1 for re-reviewing our revised manuscript. We are glad to hear that we satisfactorily addressed the comments and concerns.

Reviewer 2:

Comment:

The authors addressed all my criticism and added a significant amount of data and more appropriate data analysis. Thus the manuscript is greatly improved and suitable for publication. In my opinion the methodology will be appreciated by many scientists in different areas of research. The results concerning the ATPase cycle of Hsp90 are highly interesting as several conformational changes are detected at the single molecule level at the same time.

I have only a single final minor comment:

Figure 5 C and D: The fit lines for the closure are shown in gray and it is difficult to associate the fit lines with the corresponding cumulative dwell time data. It would be better to distinguish the fit lines by different colors.

Response:

We thank referee #2 for re-reviewing our revised manuscript. We are glad to hear that we satisfactorily addressed the comments and concerns.

In response to the final minor comment we changed the colour of the bi-exponential fit lines in Figure 5, as well as in Supplementary Figures 9 and 10, from grey to cyan in order to improve visibility.

Reviewer 3:

Comment:

The authors significantly improved the manuscript, particularly the analysis of their data. Overall I am happy with the paper in its current form and I recommend publication.

Response:

We thank referee #3 for re-reviewing our revised manuscript. We are glad to hear that we satisfactorily addressed the comments and concerns.